# HER3 targeting with an antibody-drug conjugate bypasses resistance to anti-HER2 therapies

Lucía Gandullo-Sánchez[1], Emily Capone[2], Alberto Ocaña[3], Stefano Iacobelli[4], Gianluca Sala[2,4,*] & Atanasio Pandiella[1,**]

## Abstract

Despite impressive clinical benefit obtained with anti-HER2-targeted therapies, in advances stages, especially in the metastatic setting, HER2-positive tumors remain incurable. Therefore, it is important to develop novel strategies to fight these tumors, especially when they become resistant to available therapies. We show here that the anti-HER3 antibody–drug conjugate EV20/MMAF exerted potent antitumoral properties against several models of primary resistance and secondary resistance to common anti-HER2 available therapies, including trastuzumab, lapatinib, neratinib, and trastuzumab-emtansine. HER3 was expressed in these HER2+ breast cancer cells and knockdown experiments demonstrated that HER3 expression was required for the action of EV20/MMAF. In mice injected with trastuzumab-resistant HER2+ cells, a single dose of EV20/MMAF caused complete and long-lasting tumor regression. Mechanistically, EV20/MMAF bound to cell surface HER3 and became internalized to the lysosomes. Treatment with EV20/MMAF caused cell cycle arrest in mitosis and promoted cell death through mitotic catastrophe. These findings encourage the clinical testing of EV20/MMAF for several indications in the HER2+ cancer clinic, including situations in which HER2+ tumors become refractory to approved anti-HER2 therapies.

**Keywords** breast cancer; drug resistance; HER2; HER3

**Subject Categories** Cancer; Pharmacology & Drug Discovery

## Introduction

The HER2 transmembrane tyrosine kinase plays critical roles in human homeostasis, and its deregulation has been linked to certain types of cancer (Holbro & Hynes, 2004; Arteaga *et al*, 2011; Esparis-Ogando *et al*, 2016). In fact, elevated levels of HER2 were initially reported in a subset of breast tumors and such elevated levels correlated with poor clinical outcome (Slamon *et al*, 1987). These

studies, together with preclinical data that corroborated a pro-oncogenic role of HER2 (Di Fiore *et al*, 1987), promoted the development of agents that specifically act on this tyrosine kinase. Two types of such agents have reached the oncology clinic. On one hand, antibodies targeting the extracellular region of HER2, such as pertuzumab, trastuzumab, and its derivatives, which include trastuzumab biosimilars and trastuzumab-emtansine (T-DM1), a derivative that belongs to the antibody–drug conjugate (ADC) type of anti-tumorals (Arteaga *et al*, 2011; Burris *et al*, 2011; Garcia-Alonso *et al*, 2020). These antibodies act through a complex, yet incompletely understood set of mechanisms which include down-regulation of HER2 and its signaling (Lane *et al*, 2000), inhibition of dimer formation in the case of pertuzumab (Agus *et al*, 2002), or recruitment of the immune system to sites of antibody abundance, triggering antibody-dependent cellular cytotoxicity (Clynes *et al*, 2000). In the case of trastuzumab-emtansine, internalization and release of the cytotoxic payload appear critical for its action (Burris *et al*, 2011; Garcia-Alonso *et al*, 2018). The second group of clinically approved agents that target HER2 is the small molecule membrane-permeable kinase inhibitors (TKIs) neratinib and lapatinib (Geyer *et al*, 2006; Zhang *et al*, 2009; Chan *et al*, 2016). Despite the clinical efficacy, patients with advanced disease often become refractory to the available anti-HER2 drugs (Pernas & Tolaney, 2019). Therefore, new strategies are required to overcome resistance to those therapies.

Resistance to anti-HER2 therapies may be caused by various mechanisms, including lack or truncation of HER2 (Parra-Palau *et al*, 2014), activation of downstream signaling routes, such as the PI3K pathway (Majewski *et al*, 2015), or compensatory activation of cell proliferation signals mediated by other receptor tyrosine kinases (Shattuck *et al*, 2008; Schwarz *et al*, 2017). In this respect, particular attention has been devoted to the related ErbB family receptor HER3 (Baselga & Swain, 2009). In fact, HER3 has been reported to play a pathophysiological role in resistance to anti-HER therapies (Sergina *et al*, 2007; Narayan *et al*, 2009; Garrett *et al*, 2013). Moreover, expression of several ligands of HER receptors causes resistance to trastuzumab by augmenting dimerization of HER2 with either HER3 or EGFR (Motoyama *et al*,

1  Instituto de Biología Molecular y Celular del Cáncer, CSIC, IBSAL and CIBERONC, Salamanca, Spain
2  Department of Medical, Oral and Biotechnological Sciences, Center for Advanced Studiesand Technology (CAST), University of Chieti-Pescara, Chieti, Italy
3  Hospital Clínico San Carlos, Madrid, Spain
4  MediaPharma s.r.l, Chieti, Italy
   *Corresponding author. Tel: +39 0871 541504; E-mail: g.sala@unich.it
   **Corresponding author. Tel: +34 9232 94815; E-mail: atanasio@usal.es

2002; Ritter *et al*, 2007; Phillips *et al*, 2014). Analogously, in cells made resistant to T-DM1, increased levels of the HER3 ligand neuregulin caused resistance to that ADC (Phillips *et al*, 2014). Together, these evidences indicate that acting on HER3 may be of therapeutic value for HER2$^+$ tumors, particularly in circumstances in which these tumors become refractory to the action of approved anti-HER2 drugs. Indeed, dysregulation of HER3 trafficking, as well as the ability of this receptor to interact with other receptor tyrosine kinases to modulate the sensitivity of targeted therapeutics in different cancers, has prompted the use of anti-HER3 antibodies, both as single agent or in combination with anti-cancer drugs to overcome resistance (Chakrabarty *et al*, 2012; Abel *et al*, 2013; Ma *et al*, 2014; Capone *et al*, 2015; Gaborit *et al*, 2016; Black *et al*, 2019). Several naked antibodies have been developed and tested in clinical trials. However, this approach turned out to have modest therapeutic activity (Jacob *et al*, 2018; Mishra *et al*, 2018; Liu *et al*, 2019).

Here, we report the anti-tumoral action of the HER3 antibody–drug conjugate EV20/MMAF (Capone *et al*, 2018). This ADC consists of the humanized EV20 anti-HER3 antibody (Sala *et al*, 2012, 2013; Prasetyanti *et al*, 2015) coupled with a non-cleavable linker to monomethyl auristatin F (MMAF), a synthetic compound that blocks tubulin polymerization. This antibody has been shown to block HER3/AKT signaling (Sala *et al*, 2012) and to be rapidly and efficiently internalized by cells expressing HER3 (Sala *et al*, 2013). Moreover, the antibody impaired ligand-induced signaling and clonogenic growth *in vitro* and tumor growth in BRAF-V600E mutant colon cancer (Prasetyanti *et al*, 2015). We show that EV20/MMAF exerts a potent and efficient anti-tumoral effect in different HER2$^+$ models including cells resistant to classical anti-HER2 therapeutics. Moreover, single-dose treatment with EV20/MMAF of mice implanted with HER2$^+$ cells made resistant to trastuzumab, resulted in long-lasting complete remission of tumors generated by these cells. The results encourage the clinical study of EV20/MMAF to treat HER2$^+$ tumors.

## Results

### EV20/MMAF is active in trastuzumab-resistant cells

The principal aim of this study was to explore the value of EV20/MMAF to overcome resistance to anti-HER2 drugs used in the clinic. Since trastuzumab-based therapies represent the gold standard for the therapy of patients with HER2$^+$ breast tumors in the neoadjuvant as well as the adjuvant settings (Ocana *et al*, 2018), we started the study focusing on resistance to trastuzumab. Several models of primary or secondary resistance to trastuzumab were used. One of the models of secondary resistance was created by continuous exposure of the HER2$^+$ human cell line BT474 to 50 nM trastuzumab (Fig 1A). These cells, which are sensitive to trastuzumab (Schwarz *et al*, 2017), represent a preclinical model that has been prototypically used to analyze the biology of HER2, including therapies that act on that receptor. BT474 cells chronically treated with trastuzumab (50 nM) resulted in the selection of cells that survived such treatment (Rios-Luci *et al*, 2020). Under such drug pressure, pools of cells, termed BTRH from BT474 Resistant to Herceptin™, or single clones were isolated and their response to the drug compared to that

of wild-type BT474 cells. While proliferation of naïve BT474 cells was sensitive to trastuzumab (Figs 1B and EV1A), little effect of the drug was observed in the pooled BTRH cells or in a representative resistant clone (BTRH#10). Such resistance to the action of trastuzumab was not caused by loss of total or phosphorylated (Fig 1C) or cell surface (Figs 1D and EV1B) HER2.

The effect of EV20/MMAF on BT474, BTRH, and BTRH#10 cells, which present analogous levels of total and phosphorylated and cell surface HER3 (Figs 1C and D, and EV1B and C), was then evaluated. Dose–response studies demonstrated that the drug potently affected the proliferation of these cells, assessed by cell counting after 120 h of treatment (Fig 1E). The anti-proliferative action of EV20/MMAF reached a maximum inhibitory effect at 1 nM. We next explored the sensitivity of these cells to the separate components used to generate EV20/MMAF, i.e., the EV20 anti-HER3 antibody or the cytotoxic drug MMAF. In contrast to EV20/MMAF, the nude antibody EV20 did not affect the proliferation of BT474, BTRH, or BTRH#10 cells (Fig 1F). In the case of free MMAF, the drug inhibited the proliferation of BT474 and BTRH cells (Fig 1G). However, the potency of MMAF was around three orders of magnitude lower than that of EV20/MMAF.

We complemented the above studies by using another model of secondary resistance to trastuzumab derived from a PDX (PDX118) initially sensitive to trastuzumab (Rios-Luci *et al*, 2020). To that end, a PDX118 tumor grown in a mouse was dissected and cells disaggregated and placed in culture. These cultures were propagated in the absence or presence of trastuzumab. Two clones growing in the presence of trastuzumab (named TR1 and TR2) were isolated and tested for their sensitivity to trastuzumab. As shown in Fig 2A, parental PDX118 cells were highly sensitive to trastuzumab. In contrast, clones TR1 and TR2 presented substantial resistance to the action of the drug. TR1 and TR2 clones expressed levels of HER2 similar to those of the parental PDX118 cells (Fig 2B), excluding changes in HER2 as the cause of their resistance to trastuzumab. Additional analyses indicated that all the PDX118-derived cell lines expressed similar levels of total and cell surface HER3 and HER2, independently of their sensitivity to trastuzumab (Figs 2B and EV1D). Moreover, their HER3 levels paralleled those of BT474 and BTRH cells. EV20/MMAF exerted a strong anti-proliferative effect, being equipotent in PDX118, TR1, and TR2 as well as in BT474 and BTRH cells (Fig 2C). Levels of pHER3 were higher in PDX118 cells as compared to TR1 or TR2 (Fig 2B).

With the aim of analyzing whether cells with *de novo* resistance to trastuzumab were also sensitive to EV20/MMAF, we explored the effect of trastuzumab on several human HER2$^+$ cells. The criteria for sensitivity or resistance to trastuzumab were established from the responses of BT474 and BTRH cells to the drug (Fig 1A and B). As shown in Fig 2D, SKBR3 cells responded to trastuzumab similarly to wild-type BT474 cells. On the other side, MDA-MB-361, HCC1419, HCC1569, and HCC1954 had a response to trastuzumab similar to that of BTRH cells and were therefore considered resistant cells. All the cell lines expressed HER3, in addition to HER2 (Fig 2E), confirming their reported HER2 positivity (Neve *et al*, 2006). Dose–response analyses showed that all the cell lines responded to EV20/MMAF (Fig 2F), but not to the nude EV20 antibody (Fig EV1E). The nude EV20 antibody, however, reduced the levels of HER3 and/or pHER3 in several of the treated cell lines (Fig EV1F and G).

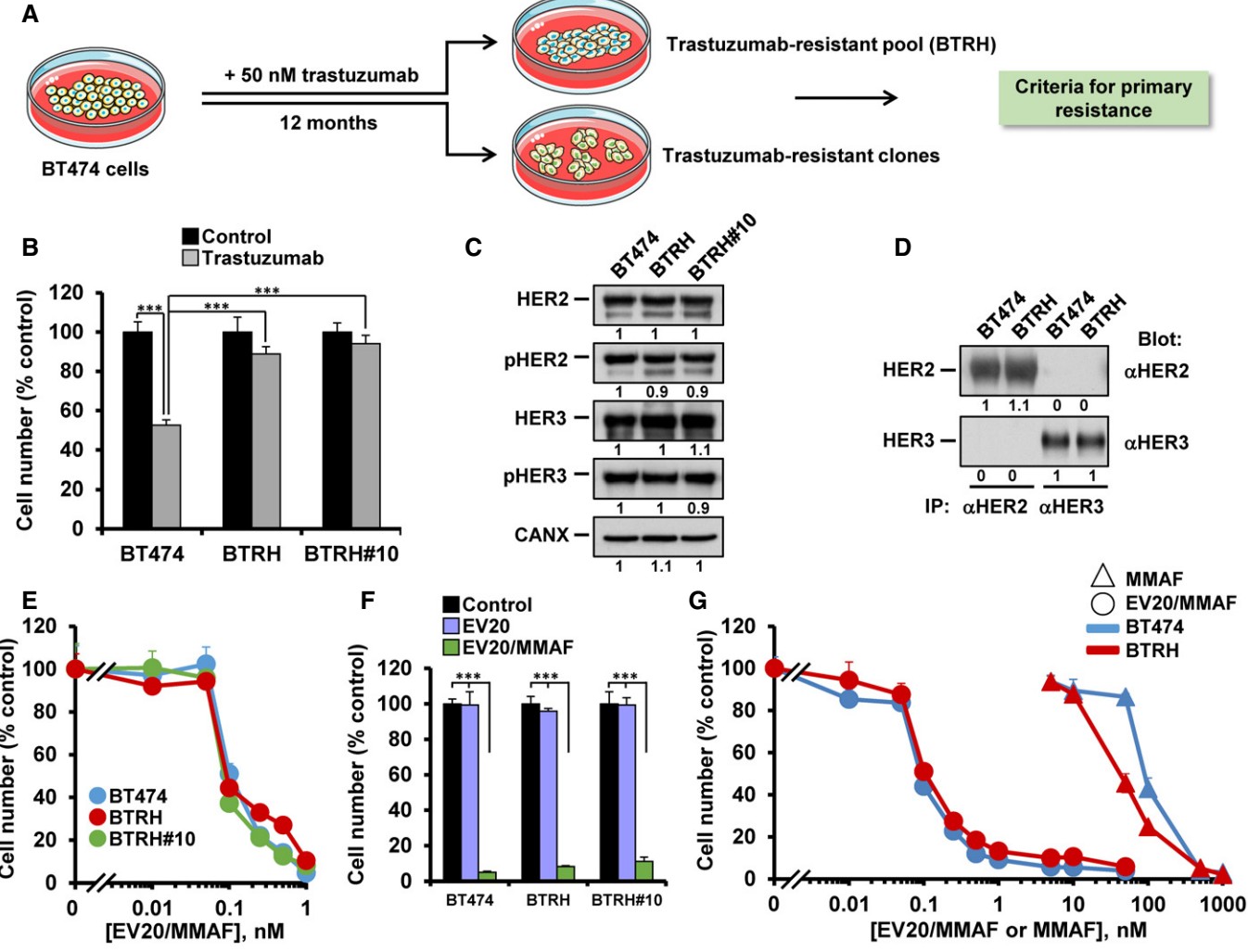

**Figure 1. EV20/MMAF efficiently acts on naïve and trastuzumab-resistant cells.**

A  Schematic representation of the generation of trastuzumab-resistant cell lines. Low-density cultures of BT474 (5,000 cells/150 mm dish) were incubated with trastuzumab (50 nM, replenished weekly) for 12 months and resistant pools and clones selected.

B  Cell counting experiments analyzing the response to trastuzumab (50 nM, 5 days) of naïve BT474 as well as the resistant BTRH pool and the BTRH#10 clone. ***$P < 0.001$. Exact $P$-values of all comparisons and the statistical test used are indicated in Appendix Table S2.

C  Western blot analyses of the expression of HER2, pHER2, HER3, and pHER3 in BT474, BTRH, and BTRH#10 cells. HER2 and HER3 were immunoprecipitated from 100 or 500 μg of cell extracts, respectively, and blots probed with anti-HER2, anti-HER3, or anti-phosphotyrosine. Calnexin was used as a loading control. Numbers below the blots show the quantitation of the signal of each band, referred to BT474 cells (intensity of band in BT474 cells = 1).

D  Analyses of cell surface levels of HER2 and HER3 in BT474 and BTRH cells by surface immunoprecipitation.

E  Response of BT474, BTRH, and BTRH#10 to EV20/MMAF. Cells treated with the indicated doses of EV20/MMAF for 5 days were counted.

F  Action of the nude EV20 antibody or EV20/MMAF (10 nM each, 5 days) on BT474, BTRH, and BTRH#10 cells. ***$P < 0.001$. Exact $P$-values of all comparisons and the statistical test used are indicated in Appendix Table S3.

G  Comparison of the effect of MMAF and EV20/MMAF on the proliferation of BT474 and BTRH cells after 5 days of treatment with the drugs.

Data information: Data in (B, E, F, and G) represent mean + SD of triplicates of an experiment that was repeated at least twice and normalized to untreated controls. In (B, E, F, and G), when error bars are invisible, that is because they are small and are covered by the graphic's symbols.

Since the presence of the HER3 ligand neuregulin has been shown to impair the action of other ADCs such as T-DM1 (Phillips *et al*, 2014), the impact of that ligand on the anti-proliferative effect of EV20/MMAF was analyzed. As shown in Fig EV1H, incubation with neuregulin slightly affected the anti-proliferative action of EV20/MMAF, although most of the action of the ADC was preserved.

## The anti-proliferative action of EV20/MMAF depends on HER3 expression levels

Whether expression of HER3 was required for the action of EV20/MMAF was then evaluated. Linear regression studies showed correlation between the levels of HER3 and the response to EV20/MMAF ($R^2 = 0.677$, $P = 0.044$, Fig EV2A and B), suggesting that the

amount of the receptor is one of the factors dictating sensitivity to the drug. In fact, the antibody had a poor effect on the human stromal cell line HS5 (Fig EV2C), a non-tumoral non-breast cell line that expresses very low levels of HER3 or pHER3 (Fig EV2D). Similarly, the antibody did not affect the proliferation of MDA-MB-231 and BT549, two triple-negative breast cancer cell lines with very low HER3 or pHER3 (Fig EV2C and D). No correlation was found between the levels of pHER3 and the potency of EV20/MMAF ($R^2 = 0.135$, $P = 0.474$, Fig EV2E and F).

To explore further the dependency of the action of EV20/MMAF on HER3 expression levels, knockdown experiments were carried out in BT474 cells. For these experiments, we selected two sequences that interfered with the expression of HER3, which also decreased pHER3 and pAKT levels (Fig 2G). The knockdown of

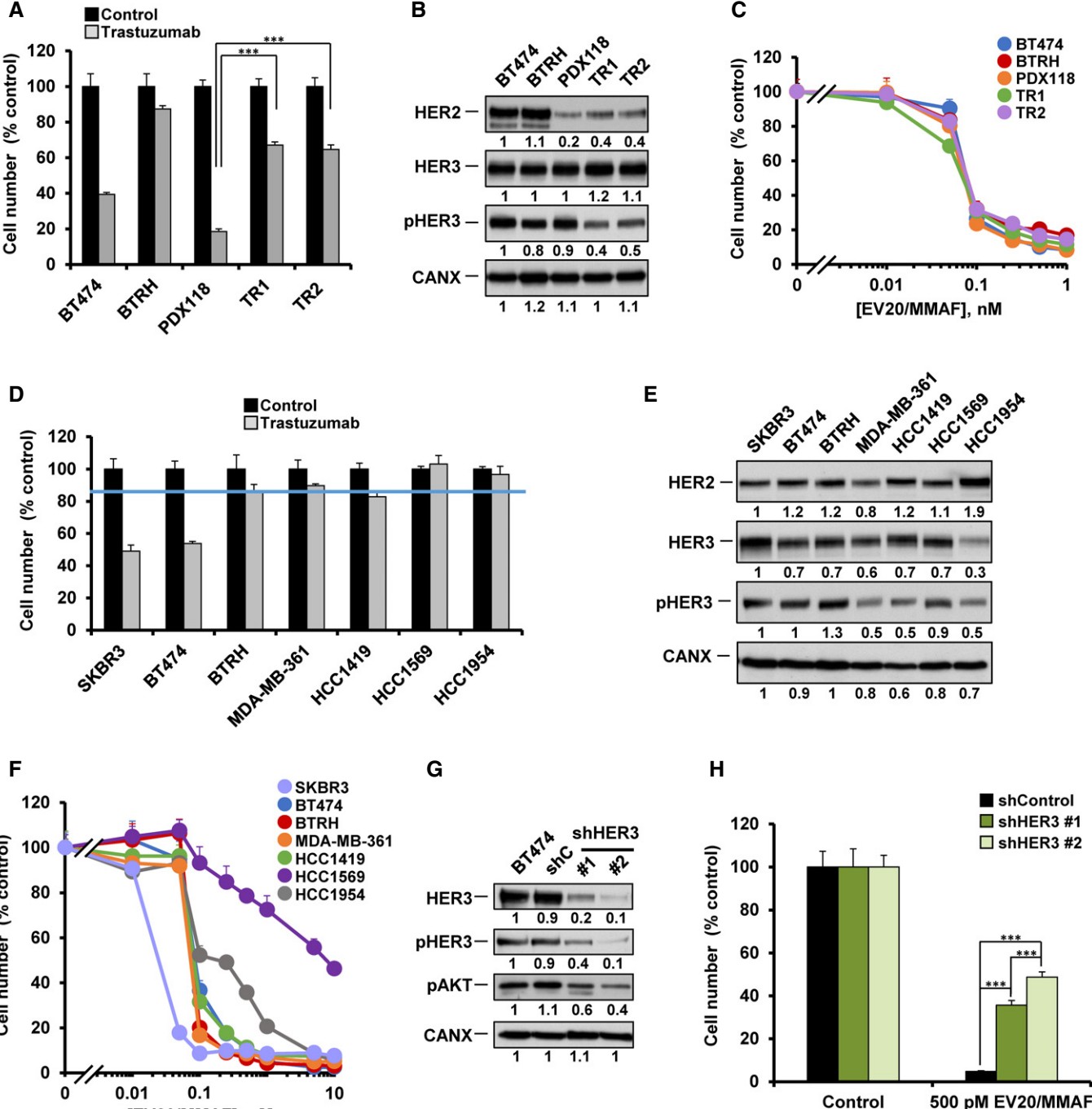

Figure 2.

**Figure 2. Action of EV20/MMAF on various models of primary and secondary resistance to trastuzumab.**

A Response to trastuzumab (50 nM, 5 days) of BT474, BTRH, patient-derived cells (PDX118), and their trastuzumab-resistant derivatives TR1 and TR2.***$P < 0.001$. Exact $P$-values of all comparisons and the statistical test used are indicated in Appendix Table S4.

B HER2, HER3, and pHER3 levels in BT474, BTRH, PDX118, TR1, and TR2. pHER3 levels were analyzed by a specific anti-pHER3 antibody.

C Effect of EV20/MMAF on BT474, BTRH, PDX118, TR1, and TR2. Cells were counted after 5 days of treatment with the indicated doses of EV20/MMAF.

D Effect of trastuzumab (50 nM, 5 days) in HER2-positive breast cancer cell lines (SKBR3, BT474, BTRH, MDA-MB-361, HCC1419, HCC1569, and HCC1954). The blue line indicates the threshold (established from BTRH) used to consider a cell line resistant to trastuzumab.

E Analyses of HER2, HER3, and pHER3 in the cells used in (D). Numbers below the blots show the quantitation of the signal of each band, referred to SKBR3 cells (intensity of band in SKBR3 cells = 1). pHER3 levels were analyzed by a specific anti-pHER3 antibody. Note that the pHER3 blot is the same as the first one showed in EV2E.

F Effect of EV20/MMAF on the cell lines used in (D). Cells were counted after 5 days of treatment with the indicated doses of EV20/MMAF.

G Knockdown of HER3 in BT474 cells. Control and HER3-targeted short hairpin sequences were transduced in BT474 cells and HER3, pHER3, and pAKT levels analyzed by Western. Levels of pHER3 were analyzed by a specific anti-pHER3 antibody.

H Effect of HER3 knockdown on the anti-proliferative action of EV20/MMAF analyzed by cell counting after 5 days of treatment with the ADC. ***$P < 0.001$. Exact $P$-values and the statistical test used are indicated in Appendix Table S5.

Data information: Data in (A, C, D, F, and H) represent mean + SD of triplicates of an experiment that was repeated at least twice and normalized to untreated controls. In (A, C, D, F, and H), when error bars are invisible, that is because they are covered by the graphic's symbols.
Source data are available online for this figure.

HER3 caused by both sequences had an inhibitory effect on the anti-proliferative action of EV20/MMAF (Fig 2H).

**EV20/MMAF is internalized in naïve and trastuzumab-resistant cells**

Immunofluorescence experiments showed that EV20/MMAF stained the cell surface at early incubation times and that staining evolved into a dotted pattern with time (Fig 3A). The accumulation of EV20/MMAF in those dots was already observed at early times (30 min) but was more clearly evidenced at 3 h or beyond. The presence of such dotted pattern was suggestive of internalization of the antibody. To verify whether EV20/MMAF was in fact internalized, the antibody was labeled with a cleavable biotin form that is sensitive to the reducing agent glutathione (GSH). If the antibody is exposed at the extracellular medium, the biotin label can be released by GSH (Esparis-Ogando *et al*, 1994). However, internalized biotinylated EV20/MMAF is protected from the reducing action of GSH, rendering biotinylated EV20/MMAF detectable by Western blotting. Internalization of biotinylated EV20/MMAF in BT474 and BTRH cells was detected at the earliest time point analyzed and reached a maximum at 60 min (Fig 3B and C). The biotinylated antibody signal decreased thereafter, indicating recycling or degradation of the internalized antibody. In both cell lines, internalization kinetics were similar.

Internalization of EV20/MMAF is expected to drive the ADC-HER3 complex to the lysosomes, where acidic proteases may cleave the ADC. To verify that EV20/MMAF reached such an acidic environment, the ADC was labeled with pHrodo, a red fluorogenic dye that is almost non-fluorescent at neutral pH, but which fluoresces in acidic environments (Nath *et al*, 2016). Cells were incubated with 10 nM pHrodo-EV20/MMAF for 24 h, and fluorescent signals analyzed by *in vivo* microscopy. In BT474 and BTRH cells, fluorescence progressively accumulated intracellularly (Movie EV1 and EV2), demonstrating that pHrodo-EV20/MMAF reached acidic compartments. Moreover, complementary immunofluorescence studies showed colocalization of EV20/MMAF with the lysosomal marker LAMP-1 (Figs 3D and EV3A and B). Finally, to confirm that arrival of the ADC-HER3 complex to the lysosomes promoted its

degradation, HER3 levels were analyzed after different treatment times with EV20/MMAF. In BT474 and BTRH cells, treatment with EV20/MMAF caused a decrease in total HER3, which was detectable between 1 and 3 h (Fig 3E and F). In contrast, the antibody did not affect the total amount of HER2. Parallel experiments performed with trastuzumab showed that this antibody did not significantly affect the levels of HER2 or HER3 (Fig EV3C and D).

**EV20/MMAF action involves cell cycle arrest and apoptosis**

To gain insights into the anti-tumoral mechanism of action of EV20/MMAF, whether such action involved a decrease in cell cycle progression, augmented cell death, or both was explored. Cell cycle assessment using propidium iodide staining revealed that EV20/MMAF increased the proportion of cells in the G2/M region of the histograms, and such increase was accompanied by a concomitant decrease in the G1 phase (Fig 4A). These changes in the cell cycle pattern caused by EV20/MMAF were similar in both cell lines. Western blotting analyses showed that EV20/MMAF caused a substantial and persistent accumulation of pHistone H3, which is used as a marker of cells in mitosis (Fig 4B). Moreover, the drug also increased the levels of pBubR1, another protein whose phosphorylation marks cells in that cell cycle phase. These Western studies also confirmed a decrease in the levels of HER3 and pHER3 upon continued treatment with EV20/MMAF in both cell lines.

The above data were consistent with induction of mitotic arrest caused by EV20/MMAF. Stalling of cells in the M phase of the cell cycle by cytotoxic payloads has been reported to provoke cell death through mitotic catastrophe (Garcia-Alonso *et al*, 2018). Such effect is caused by the action of drugs on the microtubular system required for adequate cell cycle progression. Immunofluorescence analyses showed that mitotic spindles of BT474 and BTRH cells treated with EV20/MMAF offered an abnormal aspect (Fig 4C and D). Moreover, accumulation of death cells was observed over time in cultures of cells treated with the drug (Fig 4E). Biochemical studies confirmed the induction of apoptotic cell death by EV20/MMAF since it augmented cleavage (Fig 4F) and activation (Fig 4G) of caspase 3 in BT474 and BTRH cells.

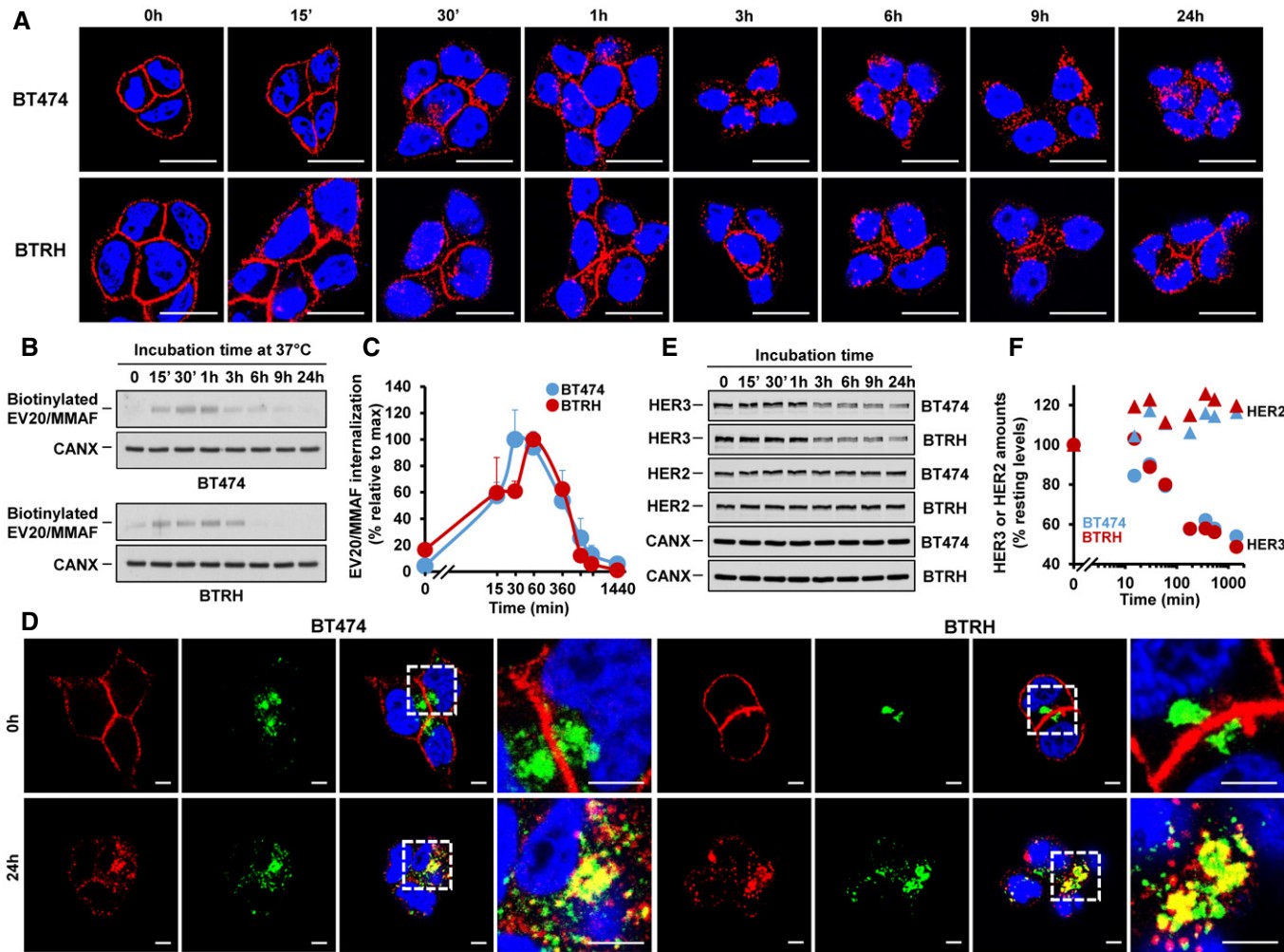

**Figure 3. Internalization of EV20/MMAF in BT474 and BTRH.**

A  Internalization of EV20/MMAF (10 nM) at indicated times in BT474 and BTRH cells. Scale bar: 20 μm. HER3 staining: red; DAPI: blue.

B  Internalization of biotin-S-S-EV20/MMAF in BT474 and BTRH cells. These cell lines were pulsed with 10 nM of biotin-S-S-EV20/MMAF for 1 h at 4°C and chased at 37°C at the indicated times. Subsequently, surface accessible biotin was cleaved, cells were lysed, and extracts precipitated with streptavidin-Sepharose. EV20/MMAF was analyzed by Western blotting using anti-human-HRP.

C  Quantitative plot of the Western studies shown in (B). Data are presented as the mean + SD of two independent experiments.

D  Colocalization (yellow) between EV20/MMAF and LAMP1 at the indicated incubation times with EV20/MMAF in BT474 and BTRH cells. Scale bar: 5 μm. HER3 staining: red; LAMP1: green, DAPI: blue.

E  Western blot analyses of the effect of EV20/MMAF (10 nM) on the levels of HER2 and HER3 in BT474 and BTRH cells at the indicated incubation times with the ADC. The same Western was probed for HER2 or HER3 expression using two differently labeled species-specific secondary fluorescent antibodies.

F  Quantitative representation of the Western studies shown in (E).

Source data are available online for this figure.

## *In vivo* anti-tumoral action of EV20/MMAF

The *in vivo* anti-tumoral effect of EV20/MMAF on BTRH cells was analyzed by orthotopically implanting these cells in the caudal mammary fat pads of female nude mice. Once tumors reached 500 mm³, mice were randomized into four groups: (i) control, (ii) EV20 (10 mg/kg), (iii) EV20/MMAF (3.3 mg/kg), and (iv) EV20/MMAF (10 mg/kg). A single intraperitoneal injection with each treatment was performed and tumor size monitored twice a week. As shown in Fig 5A, treatment with the nude antibody EV20 was unable to prevent tumor growth. In contrast, treatment of mice with

EV20/MMAF provoked tumor shrinkage, at either 3.3 or 10 mg/kg. In mice-bearing tumors created by injection of BTRH cells, weekly *in vivo* treatment with trastuzumab initially provoked tumor shrinkage, followed by tumor relapse. Because of the initial sensitivity to trastuzumab of BTRH-derived tumors *in vivo*, we also explored the *in vivo* anti-tumoral effect of EV20/MMAF on two additional resistant models. One of them, created by injecting BTRH#10 cells, represented an alternative secondary resistance model to trastuzumab, while the other, created by injecting HCC1954 cells, represented a primary resistance model. Both models showed *in vivo* resistance to weekly administered trastuzumab (Fig 5B and C). Tumors in mice

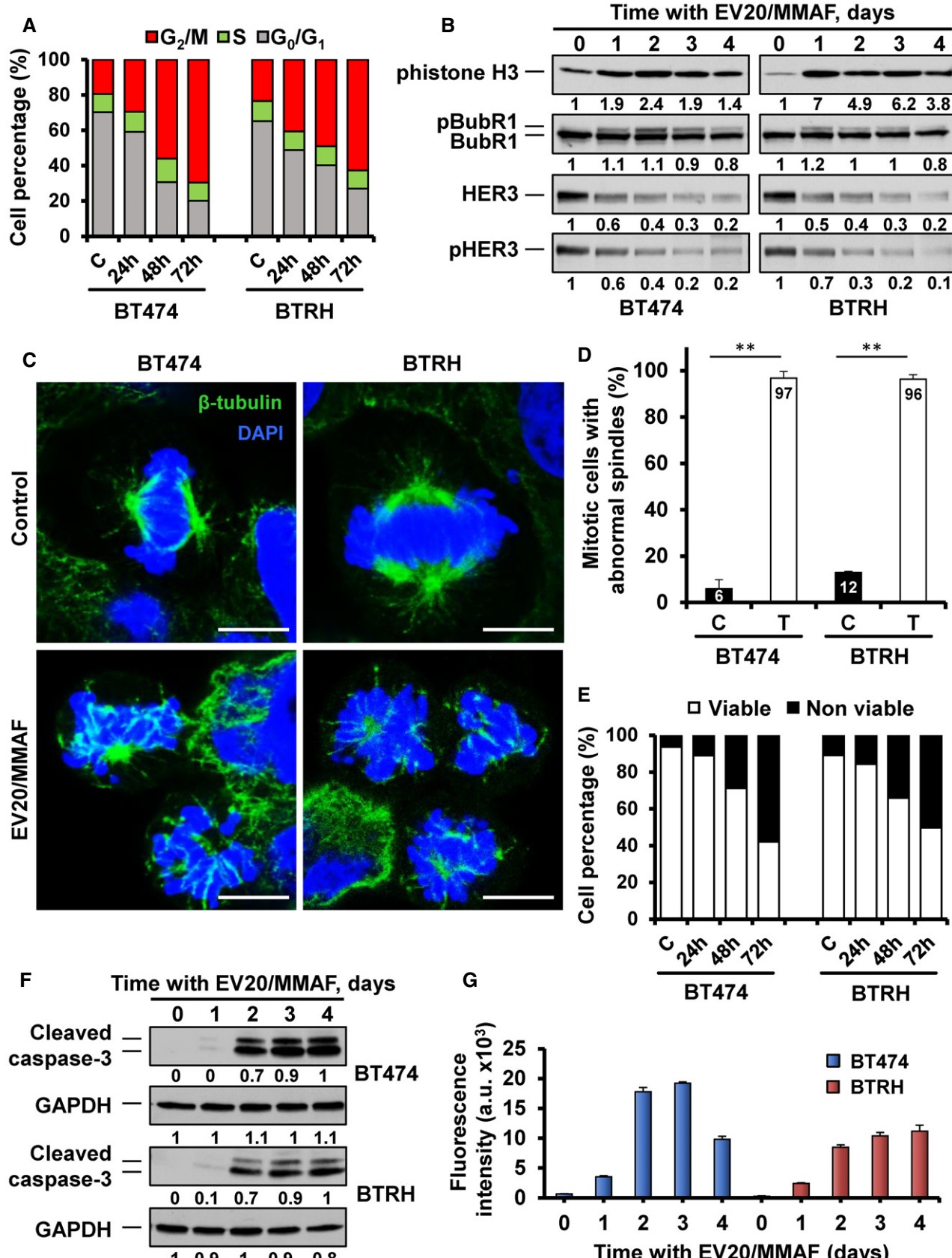

**Figure 4.**

◄

**Figure 4.  Mechanism of action of EV20/MMAF.**

A  Cell cycle analysis by flow cytometry of BT474 and BTRH cells (50,000 events) treated with EV20/MMAF (10 nM) for the indicated times.

B  BT474 and BTRH cells were treated for the indicated days with EV20/MMAF (10 nM) and lysed and the amount of different proteins analyzed by Western. Numbers below the blots show the quantitation of the signal of each band, referred to time 0.

C  Action of EV20/MMAF on mitotic spindle formation. BT474 and BTRH cells were treated (10 nM, 48 h), fixed, and stained for β-tubulin (green) and DAPI (blue). Scale bar: 7.5 μm.

D  Quantitative analysis of mitotic cells with abnormal spindles treated or not with EV20/MMAF (10 nM, 48 h) in BT474 and BTRH cells. Bars represent the mean + SD of two independent experiments, calculated as follows: (number of mitotic cells with abnormal spindles/total number of mitotic cells) × 100 (%). **$P$ < 0.01. Exact $P$-values and the statistical test used are indicated in Appendix Table S6.

E  Cell death induction by EV20/MMAF (10 nM) analyzed by double Annexin V and PI staining in BT474 and BTRH cells (50,000 events).

F  Time–course analysis of the effect of EV20/MMAF (10 nM) on cleaved caspase-3, analyzed by Western blotting in BT474 and BTRH cells. GAPDH was used as a loading control. Samples with intensity values of 1 were used as reference.

G  Fluorimetric analyses of caspase-3 activity in BT474 and BTRH cells treated with 10 nM EV20/MMAF. Data show the mean + SD of triplicates of an experiment that was repeated twice.

injected with BTRH#10 or HCC1954 cells were sensitive to the action of single-dose EV20/MMAF, confirming the effectiveness of that drug in causing tumor regression. EV20 did not affect the growth of tumors in mice injected with BTRH#10 cells, in line with the lack of effect of this antibody observed in mice injected with other BT474-derived models such as BTRH (Fig 5A) or BT-TDM1R#6 cells (Fig EV5D). In HCC1954 cells, EV20 caused tumor growth stalling, but no regression of tumors was observed (Fig 5C).

Some mice were euthanized to assess the tissue distribution of EV20/MMAF. To that end, untreated mice-bearing BTRH tumors were treated with EV20/MMAF and euthanized after 24 h or after 2 weeks. Twenty-four hours after the initial injection, EV20/MMAF could be detected in all the 11 tissues explored (Fig EV4A). The levels of the antibody were higher in lung, kidney, heart, skin, and ovary than in the tumor derived from BTRH. However, 2 weeks after the initial injection, EV20/MMAF mainly accumulated in the tumoral tissue, and its presence in most other tissues was lower or undetectable (Fig EV4B). Cleaved caspase-3 and pHistone H3 accumulated in tumors treated for 2 weeks with EV20/MMAF (Fig 5D). Total HER3 levels were lower in the samples of tumors processed after 2 weeks as compared to the samples harvested 24 h after the treatment. Different remodeling of the tissue microenvironment in both sets of samples may underlie such differences. Tumors treated with EV20/MMAF for 2 weeks presented lower levels of HER3 and pHER3 as compared to the levels present in untreated tumors (Fig 5D).

Given the tumor regression observed after treatment with a single dose of the EV20/MMAF antibody (Fig 5A), we decided to keep some of the mice under observation to explore whether tumors relapsed over time. All mice treated with a single dose of 10 mg/kg and followed along 1 year did not show recurrences of their initial tumors (Fig 5E). In the case of mice treated with 3.3 mg/kg, three out of five tumors relapsed. In these mice, new injections of EV20/MMAF were able to reduce tumor masses (Fig 5E). Having observed relapses in mice treated with a single dose of EV20/MMAF at 3.3 mg/kg, an alternative dosing scheme was followed in attempting to reduce the likelihood of such relapses. To that end, mice-bearing BTRH tumors were treated for several months with 3.3 mg/kg of EV20/MMAF, administered every 3 weeks. As shown in Fig EV4C, this treatment schedule resulted in full regression of the tumors, without relapses. Doses of 3.3 or 10 mg/kg of EV20/MMAF did not substantially affect the weight of the animals, suggesting that the drug was well tolerated (Fig EV4D). However, it should be pointed out that EV20 does not interact with murine HER3 (Capone *et al*,

2017), leaving open the possibility of potential secondary effects of EV20/MMAF due to the targeting of autochthonous HER3.

## EV20/MMAF is active in cells with acquired resistance to other anti-HER2 drugs

The action of EV20/MMAF on models of acquired resistance to other anti-HER2 drugs was also explored. To that end, we generated BT474 cells resistant to lapatinib, neratinib, or T-DM1 (Fig EV5A–C). These other resistant models were prepared following a protocol similar to that illustrated in Fig 1A, using doses of 1 or 5 μM of lapatinib, 10 nM of neratinib, or 5 nM of T-DM1 (Rios-Luci *et al*, 2017). After 3 months of treatment, pools or single clones of resistant cells were prepared and tested for their *in vitro* sensitivity to these drugs. In the case of neratinib, several clones showed sustained resistance to the drug along several months in culture. Treatment of two of these clones (BTRN#5 and BTRN#24) with EV20/MMAF had a similar effect on their proliferation as it had on the BT474 naïve cells (Fig 6A). As occurred with the other previously mentioned models of secondary resistance, the levels of HER2 and HER3 in the resistant clones were similar to those of the parental cells (Fig 6B). Levels of pHER2 and pHER3 in the neratinib-resistant clones were lower than those present in the parental cells, while those of pAKT were higher. In the case of lapatinib, the action of EV20/MMAF was tested in a pool of resistant cells (BTRL) as well as in two lapatinib-resistant clones (BTRL#3 and BTRL#109). The effect of EV20/MMAF on BTRL and BTRL#109 was similar to parental BT474 cells (Fig 6C). EV20/MMAF was slightly less potent in the BTRL#3 clone. Western blotting analyses indicated that the cells made resistant to lapatinib, either the pooled one or the clones, expressed similar levels of HER2 and HER3 and retain pHER2, pHER3, and pAKT, although at different levels (Fig 6D). Finally, we tested the efficacy of EV20/MMAF on cells resistant to T-DM1. EV20/MMAF was indeed active in these T-DM1-resistant clones, even though in clone BT-TDM1R#1, which presents deficient lysosomal processing of ADCs (Rios-Luci *et al*, 2017), the effect of EV20/MMAF was less potent (Fig 6E). HER2 and HER3 levels in T-DM1-resistant clones were similar to those of parental cells, while pHER2 levels were lower in BT-TDM1R#6 (Fig 6F). We explored the *in vivo* action of EV20/MMAF on tumors created by injecting mice with BT-TDM1R#6 cells. As shown in Fig EV5D, T-DM1 was unable to cause regression of those tumors, in agreement with previously reported data (Rios-Luci *et al*, 2017). In contrast, EV20/MMAF caused tumor regression similar to the effect observed in tumors derived from

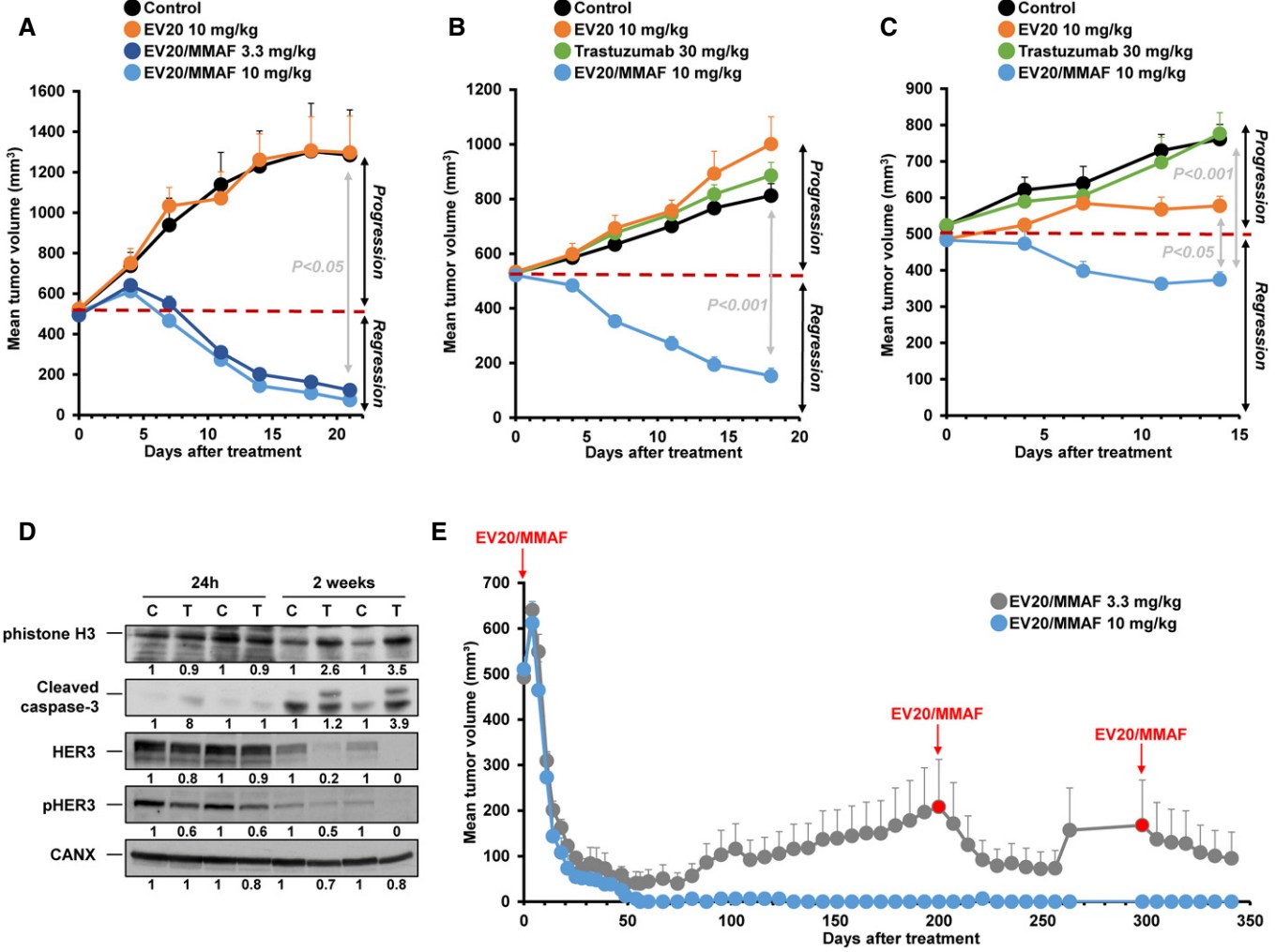

**Figure 5. *In vivo* effect of EV20/MMAF.**

A   *In vivo* effect of EV20 (10 mg/kg, *n* = 4), untreated control (PBS, *n* = 4), and EV20/MMAF (3.3 mg/kg, *n* = 5; 10 mg/kg, *n* = 4) on tumor growth in mice injected with BTRH cells. Mice were treated when tumors reached a volume of 500 mm³. Data are represented as mean + SEM. Exact *P*-values of all comparisons of the last measurements and the statistical test used are indicated in Appendix Table S7.

B   *In vivo* effect of single-dose EV20 (10 mg/kg, *n* = 3) untreated control (PBS, *n* = 4), single-dose EV20/MMAF (10 mg/kg, *n* = 4), or weekly trastuzumab (30 mg/kg, *n* = 4) on tumor growth in mice injected with BTRH#10 cells. Data are represented as mean + SEM. Exact *P*-values of all comparisons of the last measurements and the statistical test used are indicated in Appendix Table S8.

C   *In vivo* effect of single-dose EV20 (10 mg/kg, *n* = 4), untreated control (PBS, *n* = 4), single-dose EV20/MMAF (10 mg/kg, *n* = 4), or weekly trastuzumab (30 mg/kg, *n* = 4) on tumor growth in mice injected with HCC1954 cells. Data are represented as mean + SEM. Exact *P*-values of all comparisons of the last measurements and the statistical test used are indicated in Appendix Table S9.

D   Analysis of pHistone H3, HER3, pHER3, and cleaved caspase-3 in tumors of mice injected with BTRH and treated (T) or not (C) with EV20/MMAF (3.3 mg/kg for 24 h or 2 weeks). Numbers below the blots show the quantitation of the signal of each band, referred to that of a control untreated tumor.

E   Long-term monitoring of mice injected with BTRH cells and treated with EV20/MMAF (3.3 mg/kg, *n* = 5; 10 mg/kg, *n* = 4). Where indicated, mice were treated with EV20/MMAF. Data were represented as mean + SEM of the different mice groups.

Data information: In (A, B, C, and E), when error bars are invisible, that is because they are small and are covered by the graphic's symbols.

BTRH and BTRH#10 cells. EV20 did not affect the growth rate of these tumors.

## Discussion

Anti-HER2-based therapies represent the gold standard choice for the treatment of patients bearing HER2⁺ breast tumors. While introduction of these therapies has substantially improved the prognostic landscape of that type of tumors, some patients may not respond to these therapies. Therefore, identification of alternative ways to fight these resistant tumors is a medical need. Some precedents indicate that targeting HER3 may be of therapeutic value for patients whose tumors become refractory to available anti-HER2 therapies. In fact, it has been reported that tumors refractory to HER-targeted therapies, either directed to HER2 or the EGFR, express elevated levels of HER3 (Sergina *et al*, 2007; Narayan *et al*, 2009; Garrett *et al*, 2011). In the recent past, HER3-targeting agents, mainly naked antibodies,

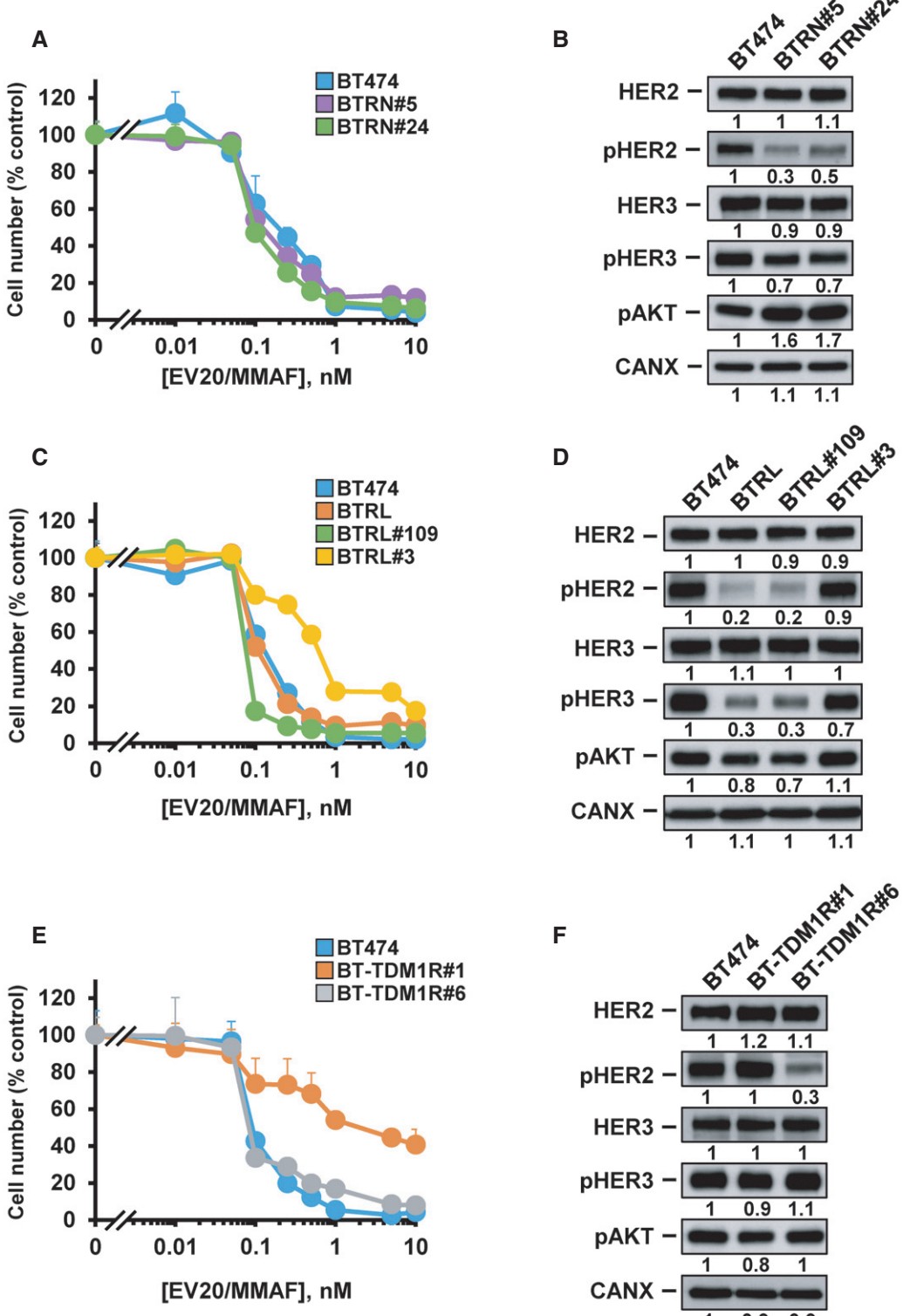

**Figure 6. Effect of EV20/MMAF in cells resistant to neratinib, lapatinib, and T-DM1.**

A–F Dose–response to EV20/MMAF after 5 days of treatment in BT474, (A) cells resistant to neratinib (BTRN#5 and BTRN#24), (C) cells resistant to lapatinib (BTRL, BTRL#109, and BTRL#3), and (E) cells resistant to T-DM1 (BT-TDM1R#1 and BT-TDM1R#6). Levels of HER2, pHER2, HER3, pHER3, pAKT, and calnexin of (B) cells resistant to neratinib, (D) cells resistant to lapatinib, and (F) cells resistant to TDM1.

Data information: In (A, C, and E), represent mean + SD of triplicates of an experiment that was repeated at least twice, and normalized to untreated controls. In (A, C, and E), when error bars are invisible, they are covered by the graphic's symbols.

have been clinically evaluated both as single agent and in combination with other targeted therapeutics. However, their therapeutic activity has proven to be limited, although tolerability was favorable (Gaborit *et al*, 2016; Jacob *et al*, 2018; Mishra *et al*, 2018; Black *et al*, 2019; Liu *et al*, 2019). More recently, we and others have provided evidence that ADCs targeting HER-3 possess potent and specific activity in metastatic melanoma as well as EGFR inhibitor-resistant lung carcinoma, pancreatic, and colorectal carcinoma (Capone *et al*, 2018; Bourillon *et al*, 2019; Hashimoto *et al*, 2019; Koganemaru *et al*, 2019; Yonesaka *et al*, 2019; Haratani *et al*, 2020). One of these novel HER3 targeting immunoconjugates is U3-1402, composed of the anti-HER3 antibody patritumab and the novel topoisomerase I inhibitor DX-8951 (Koganemaru *et al*, 2019). U3-1402 entered phase I/II study for the treatment of HER3$^+$ non-small-cell lung cancer and metastatic breast cancer. Preliminary data from these studies were encouraging, as partial responses and tumor shrinkage were observed suggesting that targeting HER3 with an ADC may be advantageous respect to targeting with naked antibody (Kogawa *et al*, 2018).

Because of these precedents and the fact that HER3 is frequently expressed in breast cancer (Vollmann-Zwerenz *et al*, 2010), we decided to explore the potential therapeutic value of the HER3 targeting ADC EV20/MMAF in HER2$^+$ breast cancer, with particular focus on the action of that drug on cells resistant to trastuzumab and other anti-HER2 therapies. EV20/MMAF had a potent anti-proliferative effect on HER2$^+$ breast cancer cells. In contrast, addition of the nude antibody EV20 did not affect the proliferation of these cells *in vitro* demonstrating that the mere action on HER3 is inefficient from the therapeutic point of view. Moreover, free MMAF had a much lower anti-proliferative action than EV20/MMAF, indicating that the sum of MMAF and EV20 optimally increases the potency with respect to the individual drugs. The *in vivo* models derived from BT474 cells confirmed the lack of effectiveness of EV20. It is relevant to comment that previous studies showing that EV20 may induce delay in tumor growth were based on a different schedule which included administration of the naked antibody twice a week for a more prolonged period (4–5 weeks; Sala *et al*, 2012, 2013). Moreover, in those studies EV20 was administered to mice-bearing tumors much smaller than the ones used in the present study (100 mm$^3$ vs. 500 mm$^3$). In the case of tumors in mice injected with HCC1954 cells, EV20 exerted a visible anti-proliferative effect. The fact that EV20 did not affect the proliferation of HCC1954 cells *in vitro*, but inhibited tumor growth of xenografted HCC1954 cells in mice, suggests that factors present in the tissue microenvironment may favor tumor growth, and they are sensitive to antibody treatment. In any case, comparison of single-dose treatment with either EV20 or EV20/MMAF demonstrated the superior *in vivo* anti-tumoral effect of the latter in all the secondary and primary anti-HER2 drug-resistant models.

The action of the antibody depended on HER3 expression as demonstrated by its lack of activity on cells naturally expressing low levels of HER3 or its decreased anti-proliferative action on cells in which expression of HER3 was genetically reduced. The correlation between HER3 levels and the anti-proliferative action of EV20/MMAF also supported the dependency on HER3. However, the numerical value of that correlation suggests that other factors, in addition to HER3, may dictate the sensitivity to EV20/MMAF. This is a relevant finding since the therapeutic use of EV20/MMAF will rely on the identification of tumors highly sensitive to the ADC. An additional factor

that may play a role in EV20/MMAF action relates to its internalization and degradation. In this respect, our immunofluorescence and biochemical analyses indicated that EV20/MMAF rapidly internalized and moved to acidic intracellular compartments, including lysosomes. In the latter, the action of lysosomal acidic proteinases is expected to generate cleaved fragments of EV20/MMAF capable of moving from the lysosomal lumen to the cytosol, where the cytotoxic payload should impair microtubule dynamics. In fact, EV20/MMAF prevented correct assembly of the microtubular machinery responsible for the progression of cells along mitosis. Such action not only affected cell cycle dynamics, but also promoted cell death through mitotic catastrophe, as occurs with other cytotoxic payloads (Montero *et al*, 2015; Garcia-Alonso *et al*, 2018). Finally, for cells in which HER3 may exert a pro-oncogenic action, its targeting with EV20/MMAF may sum to the mere ADC action as a vehicle to deliver the cytotoxic payload to HER3-expressing cells.

A relevant finding of this work was the strong anti-tumoral effect of EV20/MMAF on cells resistant to trastuzumab and other anti-HER2 therapies. Such action opens the possibility of using EV20/MMAF to fight resistance of HER2$^+$ tumors to actual anti-HER2 strategies used in the clinic. In fact, in mice implanted with cells made resistant to trastuzumab, EV20/MMAF had a strong anti-tumoral effect, with interesting characteristics. First, EV20/MMAF caused tumor regression, even from large tumors. Second, such regression was complete and long-lasting after a single dose of 10 mg/kg. Lower doses (3.3 mg/kg) also caused tumor regression but in some mice relapses were observed. However, these relapsed tumors were still sensitive to additional doses of EV20/MMAF. Furthermore, maintenance of the treatment with 3.3 mg/kg every 3 weeks for a year resulted in tumor regression in all mice used, without relapses. Those data, together with the capability to exert anti-tumoral action in cells made resistant to T-DM1, lapatinib, or neratinib and even the effect of EV20/MMAF on naïve HER2$^+$ cells offer attractive possibilities of using the anti-HER3 ADC for several indications in the HER2$^+$ cancer clinic, even beyond breast cancer. In fact, anti-HER2 therapies such as trastuzumab are used for the therapy of HER2$^+$ metastatic gastric tumors. It will be interesting to explore the value of EV20/MMAF for the therapy of these tumors. Moreover, the potent action of EV20/MMAF on HER3 expressing cells opens the possibility of extending its use for the therapy of patients bearing tumors in which HER3 expression is elevated, oncogenically mutated (Kiavue *et al*, 2020) or which may become resistant to therapies based on other HER3 ADCs (Garcia-Alonso *et al*, 2020). In fact, the development of trastuzumab-deruxtecan for patients which become refractory to previous anti-HER2 therapies, including T-DM1 (Modi *et al*, 2020), exemplifies the clinical value of using ADCs bearing different payloads but targeting the same cell surface protein. These clinical opportunities are worth being explored, once adequate biomarkers of sensitivity to the drug are available for correct patient selection.

## Materials and Methods

### Reagents and immunochemicals

Monomethyl auristatin F was from Sigma-Aldrich (Madrid, Spain). Lapatinib and neratinib were from Selleckchem (Houston, TX).

EV20 and EV20/MMAF were generated by Mediapharma and were available through Dr. Gianluca Sala. Generic chemicals were from Sigma-Aldrich, Roche Biochemicals (St Louis, USA), or Merck (Darmstadt, Germany). Trastuzumab and T-DM1 were purchased from a local pharmacy. The anti-HER3 antibody used for immunoprecipitation or Western was created in rabbits injected with a fusion protein containing a unique intracellular region of HER3 (Sanchez-Martin & Pandiella, 2012). Other antibodies used are described in Appendix Table S1.

## Cell lines and viral infections

Culture media was from GIBCO BRL (Gaithersburg, MD, USA). The cell lines were cultured either in DMEM (BT474, PDX118, and resistant cells lines derived from them, SKBR3, MDA-MB-361, MDA-MB-231, HS5, and HEK293T) or in RPMI (BT549, HCC1419, HCC1569, HCC1954) supplemented with 10% FBS and antibiotics (penicillin 100 U/ml, streptomycin 100 μg/ml). All the cell lines were obtained from the ATCC except PDX118, TR1, and TR2 which were generously provided by Dr. Joaquin Arribas laboratory (VHIO, Barcelona). Cells resistant to trastuzumab or T-DM1 were generated by continuous exposure to the respective drugs, as described (Rios-Luci *et al*, 2017, 2020; Diaz-Rodriguez *et al*, 2019). Briefly, for the generation of trastuzumab-resistant BT474 cells, 5,000 parental BT474 cells were plated in 150 mm dishes that were then treated with 50 nM trastuzumab for 12 months (media replaced weekly). Then, pools of resistant cells or individual clones were prepared and maintained in the absence or presence of trastuzumab for an additional 6-month period. The reason to maintain cultures with or without trastuzumab was to explore the stability of the resistant phenotype. In fact, we found unnecessary to maintain trastuzumab once resistance to the drug was established. Generation of T-DM1-resistant cells was achieved by culturing BT474 cells (5,000/150 mm dish) for three months in the presence of 5 nM T-DM1. Resistant clones were isolated by single-cell cloning as described (Rios-Luci *et al*, 2017). For the generation of lapatinib or neratinib-resistant cells, BT474 cells were plated at 5,000 cells/150 mm petri dishes and treated with 1, 5, 10, 50, 100, 500, 1,000, or 5,000 nM of the drugs. The best selection was achieved with doses of 1 and 5 μM of lapatinib and 10 nM of neratinib. After 3 months in culture under the continuous presence of the drug (media replaced weekly) pools or clones of resistant cells were obtained and tested for their sensitivity to those drugs, as compared to naïve BT474 cells. The action of trastuzumab or the other anti-HER2 drugs on the resistant cells was explored by cell counting or MTT assays. Knockdown of HER3 was performed by infection with lentiviral particles as described (Seoane *et al*, 2016). Briefly, 4 μg of the following plasmids: pMDLg/RRE, pRSV-Rev, and pMD2.G (Addgene, Cambridge, MA, USA), along with 8 μg of the pLKO.1 lentiviral plasmid containing a scramble shRNA (sh-Control) or the indicated shRNA (Sigma-Aldrich, St. Louis, MO, USA) were cotransfected into HEK293T cells using jetPEI® reagent (Polyplus-transfection, Illkirch, France) following the manufacturer's instructions. Twenty-four hours later, HEK293T medium was replaced with fresh medium, and 48 h after the cotransfection, the medium containing lentiviral particles was collected, filtered, and used to infect cells after the addition of 6 μg/ml polybrene. Cells were cultured for 48 h to allow for efficient protein knockdown and were subsequently selected with 3 μg/ml puromycin (Sigma-Aldrich, St. Louis, MO, USA) for another 48 h. Five different shRNA sequences targeting HER3 were tested and those two that produced higher knockdown were used.

## Western blotting and immunoprecipitation

The preparation of cell lysates, immunoprecipitation, and Western blotting procedures has been described previously (Esparis-Ogando *et al*, 2008; Seoane *et al*, 2010). To prepare cell lysates, cells were washed with phosphate-buffered saline (PBS) (137 mM NaCl, 2.7 mM KCl, 8 mM $Na_2HPO_4$, 1.5 mM $KH_2PO_4$) and lysed in ice-cold lysis buffer (20 mM Tris–HCl pH 7.0, 140 mM NaCl, 50 mM EDTA, 10% glycerol, 1% Nonidet P-40, 1 μM pepstatin, 1 μg/ml aprotinin, 1 μg/ml leupeptin, 25 mM β-glycerol phosphate, 50 mM sodium fluoride, 1 mM phenylmethyl sulfonyl fluoride, 1 mM sodium orthovanadate) for 15 min. Lysates were cleared by centrifugation at 17,000 *g* at 4°C for 10 min and supernatants transferred to new tubes. Protein concentration in the lysates was measured using the BCA Assay (Thermo Fisher Scientific, Madrid, Spain). For immunoprecipitation, equal amounts of protein were incubated with the corresponding antibody and protein A-Sepharose at 4°C for at least 2 h. Immune complexes were recovered by a short centrifugation at 17,000 *g* for at least 30 s, followed by three washes with 1 ml ice-cold lysis buffer. Samples were then boiled in electrophoresis sample buffer and resolved by sodium dodecyl sulfate–polyacrylamide gel electrophoresis (SDS–PAGE) gels. The percentage of the gel was chosen based on the molecular weight of the protein to be analyzed. For immunoblotting, proteins in gels were transferred to polyvinylidene difluoride membranes (Millipore Corporation, Bedford, MA, USA) using a wet transfer apparatus (Mini Trans-Blot Cell, Bio-Rad, Hercules, CA, USA). Membranes were then blocked in Tris-buffered saline with Tween (TBST) (20 mM Tris–pH 7.5, 150 mM NaCl, 0.1% Tween 20) containing 1% of bovine serum albumin for at least 1 h and then incubated with the corresponding antibody for 2–16 h at the dilutions indicated in Appendix Table S1. After washing with TBST, membranes were incubated with HRP-conjugated secondary antibodies for 30 min, washed three times, 7 min each time, and bands visualized by enhanced chemiluminescence. Densitometric measurements of the bands were done using ImageJ 1.44 software (National Institutes of Health, Bethesda, MD, USA) or Image Lab Software 6.0.1 (Bio-Rad). GraphpadPrism 6.0 (GraphPad Software, La Jolla, CA, USA) was used to determine $IC_{50}$ values. In experiments in which a condition was compared to a control, the intensity of the latter was taken as 1.

## Cell proliferation, cell cycle, and apoptosis analyses

Cell proliferation was analyzed by cell counting assays (Rios-Luci *et al*, 2020). Briefly, cells were seeded in 6-well plates, and 24 h later, the medium was replaced with complete medium containing the desired drugs. Cells were collected and counted using a Z1 Coulter Particle Counter (Beckman Coulter, Pasadena, CA, USA). Cell cycle and apoptosis analyses were performed as described (Montero *et al*, 2015). For both assays, cells were cultured in 6-well plates and then were treated with drugs as the indicated time. Then, cell monolayers were collected by pooling together the non-attached and attached.

For cell cycle analysis, cells were fixed in ice-cold 70% ethanol minimum overnight. Then, cells were centrifuged at 300 $g$ for 10 min and resuspended in 500 μl of PBS containing 500 μg/ml DNAse-free RNAse for 2 h at 37°C. After that, propidium iodide (5 μg/ml) was added and DNA content analyzed in an Accuri C6 Flow Cytometer (BD).

For apoptosis assays, the protocol of the FITC-Annexin V Apoptosis Detection Kit I (BD Biosciences) was followed. Attached and non-attached cells were collected and resuspended in 100 μl of ice-cold binding buffer (10 mM HEPES pH 7.4, 140 mM NaCl, 2.5 mM CaCl$_2$) containing 5 μl of Annexin V-FITC and 5 μl of PI (50 μg/ml). Samples were incubated for 15 min at room temperature, and after this time, 400 μl of binding buffer was added to each tube and cells acquired in an Accuri C6 Flow Cytometer (BD). In both experiments, 50,000 events were collected and analyzed.

To analyze caspase 3 activity, cells were treated as required and lysed in lysis buffer with protease and phosphatase inhibitors. A 50 μg of protein was incubated in the caspase reaction buffer (25 mM HEPES pH 7.4, 150 mM NaCl, 1 mM EDTA, 0.1% CHAPS, 10% sucrose), 10 mM DTT and containing 5 μM of the fluorescent substrate Ac-DEVD-AFC (BD Biosciences) at 37°C for 1 h at dark. The fluorometry signal was measured at 400/505 nm in a multi-well fluorescent reader (BioTek). Analysis of mitotic cells was carried out as described (Rios-Luci et al, 2017). In brief, cells were seeded at low density, treated with EV20/MMAF, and stained with DAPI and anti-β-tubulin. Then, cells were observed in a Zeiss Axiophot 2 microscope with 63× oil immersion objective. For each condition, at least 2,000 cells were counted. Cell with condensed chromosomes was considered mitotic cells.

## Cell surface HER2 and HER3

Cytometric assessment of cell surface HER3 was performed by incubation for 30 min with 10 nM of EV20/MMAF. The cell-bound antibodies were detected by incubation with a cyanine 3-conjugated anti-human secondary antibody. Mean fluorescence levels were determined using a BD Accuri C6 flow cytometer. For cell surface immunoprecipitation, cells were washed twice with Krebs-Ringer-HEPES buffer(140 mM NaCl, 5 mM KCl, 2 mM CaCl$_2$, 5 mM MgSO$_4$, 1.2 mM KH$_2$PO$_4$, 50 mM HEPES, pH 7.4) (Cabrera et al, 1996) and incubated in the same buffer with 10 nM EV20/MMAF or trastuzumab for 2 h at 4°C. Then, cells were lysed and cell-bound antibodies precipitated with protein A-Sepharose. Immunocomplexes were then washed and HER2 or HER3 detected by Western.

## Immunofluorescence microscopy

For immunofluorescence (IF) detection of EV20/MMAF, 100,000 or 150,000 BT474 or BTRH cells were cultured on glass coverslips inserted into 35 mm dishes and treated with 10 nM EV20/MMAF for the indicated times. Then, cells on coverslips washed with PBS/CM (1 mM CaCl$_2$, 0.5 mM MgCl$_2$ in PBS), fixed in 2% paraformaldehyde, and then washed with PBS/CM. After that, incubations were quenched with 50 mM NH$_4$Cl, and cells were permeabilized (0.1% triton, 0.2% BSA) and then coverslips with cells incubated for 1 h in blocking solution, that contained PBS/CM with 0.2% BSA. Before incubation with Cy3-conjugated anti-human antibody, cells were washed in PBS/CM/BSA. Samples were analyzed

by confocal IF microscopy using a Leica TCS SP5 system (Leica Microsystems CMS, Wetzlar, Germany). For indirect IF, cells were cultured and processed as above, including an incubation step with primary antibodies against β-tubulin or LAMP1 (Rios-Luci et al, 2017). Cy2-conjugated goat anti-mouse was used to detect β-tubulin and Cy2-conjugated goat anti-rabbit for LAMP1. Colocalization was analyzed with Leica Application Suite Advanced Fluorescence (LAS AF Version 2.7.3.9723). For live cell microscopy evaluation of EV20/MMAF internalization and targeting to acidic intracellular compartments, the antibody was labeled with pHrodoTMiFL Red Microscale Protein Labeling Kit (Thermo Scientific) according to the manufacturer's instructions and following published procedures (Rios-Luci et al, 2017). Images were acquired every 15 min for 24 h using a Nikon Eclipse TE2000-E microscope. Videos were prepared with Image J.

## Biotin labeling and internalization of EV20/MMAF

EV20/MMAF was biotinylated with EZ-Link NHS-SS-Biotin (Thermo Scientific) according to the manufacturer's instructions and as described (Rios-Luci et al, 2017). Briefly, 500 μg EV20/MMAF was incubated with a 20-fold molar excess of a 10 mM NHS-SS-Biotin solution in DMSO. The non-reacted NHS-SS-Biotin was quenched with 50 mM of NH$_4$Cl. Cells were treated with 10 nM biotin-labeled EV20/MMAF for 1 h at 4°C, then washed twice with PBS to remove any unbound biotin-EV20/MMAF, and chased for the indicated times at 37°C. Non-internalized biotin-labeled EV20/MMAF was removed by washing three times with ice-cold freshly prepared cleavage buffer (Rios-Luci et al, 2017). Cells were then washed twice with PBS and lysed in ice-cold lysis buffer (150 mM NaCl, 1 mM EDTA, 10% glycerol, 40 mM HEPES pH 7.4, 1% Nonidet P-40, and phosphatase and protease inhibitors). Biotin-labeled EV20/MMAF was precipitated with streptavidin-Sepharose high performance (GE Healthcare), for at least 2 h at 4°C. Immunocomplexes were recovered, washed, and resolved by SDS–PAGE and standard Western blotting.

## Xenograft studies

Female BALB/c nu/nu mice (8–9 weeks old) were purchased from Charles River Laboratories and maintained and manipulated at the animal facility of the University of Salamanca following legal and institutional guidelines. A total of $5 \times 10^6$ BTRH, BTRH#10, HCC1954, or BT-TDM1R#6 cells resuspended in DMEM and Matrigel (BD Biosciences) were injected orthotopically into the mammary fat pad, at two sites per mice. When tumors reached a volume of 500 mm$^3$, animals were randomized to four groups with similar mean tumor volumes that were intraperitoneally treated with vehicle alone (PBS), EV20 (10 mg/kg), T-DM1 (15 mg/kg), trastuzumab (30 mg/kg), or EV20/MMAF (3.3 mg/kg or 10 mg/kg). Tumor diameters were measured with calipers, and tumor volumes were calculated by the following formula: volume = (width$^2$ × length)/2. At the time of sacrifice, some tumor samples or tissues were collected and immediately frozen in liquid nitrogen. They were later lysed using a Dispomix drive tissue homogenizer (Medic tools), and EV20/MMAF was analyzed by immunoprecipitation of 500 μg of protein with protein A-Sepharose and Western blotting with the HRP-conjugated anti-human secondary antibody.

### The paper explained

**Problem**
HER2-targeted therapies have revolutionized the therapy of patients bearing HER2[+] tumors. Yet, some patients, especially those with advanced disease, may develop resistance to these therapies, underlining the clinical need for alternative treatment strategies.

**Results**
In this study, we show that the antibody–drug conjugate EV20/MMAF, which targets the cognate receptor HER3, exerts a potent anti-proliferative action on HER2[+] breast cancer cells resistant to the conventional anti-HER2 agents trastuzumab, lapatinib, neratinib, and the trastuzumab derivative T-DM1. *In vivo*, EV20/MMAF provoked long-lasting regression of tumors in mice implanted with cells resistant to the gold standard anti-HER2 drug trastuzumab.

**Impact**
The anti-tumoral property of EV20/MMAF in cells resistant to anti-HER2 therapies, and even on naïve HER2[+] cells, raises the attractive possibility of using EV20/MMAF in the therapy of HER2[+] breast cancer.

## Statistical analyses

In proliferation experiments, each condition was analyzed in triplicate and data presented as mean + SD of an experiment representative of at least 2 independent experiments. Statistical analyses were carried out using SPSS 25 (SPSS Inc., Chicago, IL, USA). Pearson correlation and regression analyses were used for the correlation studies. Two-way factorial ANOVA with Bonferroni correction was used when more than one factor was present. One-way ANOVA was used to compare more than two groups. The Bonferroni post hoc test or Games–Howell post hoc test was used in case of variance homogeneity or heterogeneity, respectively. Data distributions were checked for normality by the Shapiro–Wilk test, and homogeneity of variances was checked by the Levene test. Welch's *t*-test was used to compare two groups when the distribution is assumed to be normal, and the samples have unequal variances.

**Expanded View** for this article is available online.

## Acknowledgements
AP: Ministry of Economy and Competitiveness of Spain (BFU2015-71371-R), the Instituto de Salud Carlos III through the Spanish Cancer Centers Network Program (RD12/0036/0003) and CIBERONC, the Scientific Foundation of the Spanish Association Against Cancer (AECC), ALMOM, and the CRIS Cancer Foundation. Work carried out in AP and AO laboratories receives support from the European Community through the Regional Development Funding Program (FEDER). GS: Fondazione-AIRC (IG GRANT 2016, Id 18467). EC is the recipient of an AIRC fellowship. LGS is recipient of a predoctoral contract (BES-2016-077748).

## Author contributions
LGS: performed the experiments, interpreted data, read, and corrected the paper. EC: prepared the antibody, read, and corrected the paper; AO: read the paper and suggested ideas; SI: read the paper and suggested ideas; GS: designed the study and the antibody, read, and corrected the paper; AP: designed the study and wrote the paper.

## Conflict of interest
GS and SI are shareholders of Mediapharma s.r.l.; and the other authors declare that they have no conflict of interest in this study.

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
