## [Review Process File · EMBO Molecular Medicine]

HER3 targeting with an Antibody Drug Conjugate Bypasses Resistance to anti-HER2 therapies

Lucía Gandullo-Sánchez, Emily Capone, Alberto Ocaña, Stefano Iacobelli, Gianluca Sala and Atanasio Pandiella

Review timeline:

Submission date:	24th Sep 2019
Editorial Decision:	16th Oct 2019
Revision received:	4th Mar 2020
Editorial Decision:	20th Mar 2020
Revision received:	24th Mar 2020
Accepted:	29th Mar 2020

Editor: Lise Roth

Transaction Report:

1st Editorial Decision

16th Oct 2019

Thank you for the submission of your manuscript to EMBO Molecular Medicine. We have now heard back from the three referees whom we asked to evaluate your manuscript.

As you will see from the reports below, while they all mention the potential translational relevance of the findings, they also raise substantial concerns, which should be convincingly addressed in a major revision of the present manuscript.

In particular, given the somewhat compromised novelty of the manuscript, efforts should be made to:

- Characterize drug resistance in vitro,
- Improve the introduction and the discussion in light of previously published work,
- Expand the in vivo work.

However, we will not ask you to generate a mouse model of HER2/HER3 overexpression and resistance, as this would likely not be feasible in a reasonable timeframe (point 6 from referee #3).

Addressing the above-mentioned concerns, as well as the additional reviewers' points in full will be necessary for further considering the manuscript in our journal (with the exception of point 6, referee #3). We realize that revising the manuscript according to the referees' recommendations will require a lot of additional work and experimentation. I am unsure whether you will be able or willing to address those and return a revised manuscript within the 3 months deadline. On the other hand, given the potential interest of the findings, I would be willing to consider a revised manuscript with the understanding that acceptance of the manuscript would entail a second round of review. EMBO Molecular Medicine encourages a single round of revision only and therefore, acceptance or rejection of the manuscript will depend on the completeness of your responses included in the next, final version of the manuscript. For this reason, and to save you from any frustration in the end, I would strongly advise against returning an incomplete revision and would also understand your decision if you choose to rather seek rapid publication elsewhere at this stage.

***** Reviewer's comments *****

Referee #1 (Remarks for Author):

The manuscript by Gandullo-Sanchez et al entitled "HER3 targeting with an antibody drug conjugate bypasses resistance to anti-HER2 therapies" describes a new study using EV20/MMAF, an anti-HER3 drug conjugate that displayed, in preclinical setting, a potent antitumor activity in cutaneous melanoma, to demonstrate its potent anti-tumor properties in the context of HER2+ breast cancers resistant to HER2 targeting therapies such as trastuzumab, lapatinib, neratinib. The authors explored the action of EV20/MMAF on in vitro primary and secondary (acquired) resistance models and reported a long-lasting tumor growth inhibition induced by EV20/MMAF on one mice model of trastuzumab-mediated resistance. From this point of view, the topic of the paper is clearly of interest as this study strengthens the arsenal of strategies targeting HER3 with very promising results in the field of breast cancers, and more generally validates and enlarges the anti-tumor potential of EV20/MMAF.

However, an important part of the manuscript is devoted to re-confirming mechanisms or properties related to either the antibody EV20 or the drug conjugate such as the HER3 dependent cell killing activity, antibody-mediated internalization or drug-mediated cell cycle termination, that have been described in previous publications. Even if these points are supported here by new data/techniques, they only confirm data. It would have been more interesting to extend in vivo the in vitro studies on the efficacy of this ADC against different resistance models (primary resistance, drug and/or trastuzumab ADC). This would have bring even more validity and facilitate clinical translation.

The experiments are reasonably well-designed and most results are sound.

However, the manuscript needs to be improved by the following recommendations:

Specific comments

1- In the introduction, the authors give only very general information about HER2-positive breast cancers and HER3. The role of HER3 in treatment failure in cancer therapy is well known and different approaches targeting HER3 are currently ongoing (pre-clinic and clinic) in different cancers. A more precise contextualization would have been a plus, particularly with a brief overview of the current researches in the field. As well, a brief reminder of the properties of EV20, notably its capacity to block ligand dependent and independent activation of HER3 would be of help.

2- Concerning the model of secondary resistance, very little technical details are given, neither in the text nor in Fig1A that does not provide much and thus can be removed.

- What low density of cells really means (Fig.1A legend)?
- Frequency of trastuzumab replenishment during the 1-year treatment?
- Is resistance assessed only by viability assay? no phenotype investigation?
- How are the cells maintained once made resistant, just in medium without low dose of trastuzumab? Is the resistant phenotype stable? A reference or more technical details should be given.

3- The authors stated that resistance to the action of trastuzumab was not caused by a loss of phosphorylated HER2, but it is not so obvious from the western blot (Fig 1C). Addition of intensity values obtained by densitometry could be of interest. As well in the same blot, an increase of HER3 is observable but is not commented.

4- In the legend of Fig1, does the sentence "Data are represented as mean + SD of triplicate..." apply to all cell counting graphs? if not, the number of performed experiments should be added. The error bars are missing or not visible for many points. These remarks are valid for most figures.

5- In Fig2B the HER2 level is lower than that of HER3 in BT474 and BTRH while in Fig.1C it is the opposite. Is there any explanation?

6- In Fig2F, is there any explanation for the lower cytotoxic activity of EV20/MMAF on HCC1569 despite a similar level of HER3, and compared to its activity on HCC1954?

7- The paragraph (and the number of panels Fig.2G-K) on the HER3 dependent cell killing activity

of EV20/MMA (p7) should be condensed to the data considered by the authors as the most relevant and new in relation to the data already published on this subject. The other data can be moved to supplementary data.

The linear regression analysis is very briefly described, so a more descriptive figure showing a plot with all data and the linear regression analysis should be given instead of Fig.2C.

8- The sentence "For these experiment, cells were incubated...." should be removed (experimental details given in Mat &Met).

The name of the cells used in the internalization of the biotinylated ADC is not mentioned in the text, only in the figure.

9- What about the impact of HER3 ligand on the cytotoxic activity of EV20/MMAF? It would have been of interest to verify (as they did for the internalization, cell-cycle..), that the presence of neuregulin does not affect the cytotoxicity of EV20/MMAB.

10- It is not clear whether the mice to be euthanized were picked in the same group or among the 3 treated groups (p10)?

Fig.5B is of poor quality: the samples are barely visible due to truncation of the gel and the loading control varies a lot (both in amount and number of band). How was made the quantification based on this? As well, in Fig EV3A, variability of the loading control: in this respect, one could assume that the level was also high in kidney and intestine, no?

11- During the in vivo studies, do the BTRH cells remain resistant to trastuzumab? A control with trastuzumab treatment is missing and should be added to ensure the stability of the phenotype. Or at least it should be discussed.

As well, does an ADC-control be of interest?

12- The discussion is very focused on the results without much openness to the literature. It should be more detailed to highlight the advantage or/and weakness of their molecule compared to the various molecules in pre-clinical and clinical trials in progress (for exemple anti-HER3 antibody drug conjugate U3-1402?)

13- Mat et Met section should be improved so that the experiments can be reproduced because several methodological details are missing (number of cells, secondary antibodies for detection of tubulin and LAMP1 in indirect IF, number of replicate, p-Tyr antibodies which ones for HER2 and HER3? ..)

When do they use anti-EGFR antibody for immunoprecipitation?

Anti-HER3 has been described: do the authors mean the EV20?

Can the authors explain the choice not to give statistical analyses of their data?

14- The figures are very dense and could be reduced by selecting the most relevant data most of the time.

Addition of intensity values obtained by densitometry should be added to all western blot images

The error bars are often not visible in the proliferation curves

The color legend of the immunofluorescence images in Fig. 3A and D are missing in the legend.

Minor modifications:

p7 supplementary Fig1D should be replaced by Fig. EV1D

p10, Twenty instead of 24, no number at the beginning of a sentence

p11, supplementary Fig 2B should be replaced by Fig. EV3B

Referee #2 (Comments on Novelty/Model System for Author):

I rated the novelty medium since one of the results on the MOA of the ADC has already been shown for the humanized antibody. I mentioned this in the review. Otherwise the rest is novel. So it's only a small effect on novelty.

Referee #2 (Remarks for Author):

The manuscript by Gandullo-Sanchez and collaborators presents cellular and *in vivo* work on a novel HER3 targeting antibody drug conjugate (ADC), EV20/MMAF. The ADC is composed of a humanized antibody targeting HER3, EV20, conjugated to auristatin F, a tubulin polymerization-blocking compound. There are two important reasons to generate and study therapeutics targeting HER3. First, since the approval of trastuzumab for treatment of HER2 positive breast cancer, many, but not all patients with high HER2 levels have benefitted from treatment. It is well known that at least some of the trastuzumab resistance is due to activation of HER3 the main regulator of the PI3K/AKT pathway. Second, oncogenic mutations in HER3 have been found in many cancer types, most prevalently in gastric and colon cancers. The work presented by Gandullo-Sanchez et al concentrates on trastuzumab resistant models. The paper is clearly written and the presented data are interesting and well discussed. I do have a few suggestions and criticisms, which if addressed properly could make the paper even better.

General comments:

1. There are now a number of HER3 targeting therapeutics that are in clinical development. The clinicaltrials.gov site lists at least 25 trials with antibodies targeting HER3 in various types of human cancer. There are even publications describing clinical trial results for some of them, e.g., lumretuzumab. I was surprised that there was no discussion of at least some of these in the manuscript. It would be important to discuss first, why the new EX20/MMAF ADC might be better than what's already available and second, whether its MOA differs and might have some advantages compared to others.

2. Throughout the paper, the authors seem to equate HER3 levels with oncogenic activity, arguing, e.g., that the human stromal cell line HS5 has low HER3, which could explain why it is not sensitive to the antibody (Fig 2). In my opinion, they should be looking at P-HER3 levels, since this is a better sign of activity and dependency. There is one P-HER3 blot shown, but I would like to see more, particularly in the *in vivo* work in Fig 5.

3. The original murine mAb MP-RM-1 was described and studied in Sala et al 2012 *Oncogene*, and the humanized version EV20 was characterized in Sala et al 2013 *Translational Oncol*. In both these papers it was shown that the antibodies downregulate HER3 levels and AKT activity (p-AKT). Using microscopy, it was shown in Sala et al 2013 that there was a time-dependent increase in co-localization of EV20/LysoTracker and that the antibody was still bound to HER3. This work impacts somewhat on the originality of the current paper, where they also nicely show with stills and with videos that EV20/MMAF tracks to the lysosomes.

4. The final general comment/criticism that I have relates to the *in vivo* activity of EV20/MMAF. Both previous papers show that the MP-RM-1 and the EV20 antibodies block *in vivo* outgrowth of different human xenograft models. In the current manuscript, they show that "the nude antibody EV20 was unable to prevent tumor growth" (Fig 5A, pg 10). I was surprised to find that there was no discussion of the difference between their results and what has been published.

Specific comments to the figures:

Fig 2 -

Since SKBR3 and BT474 are known to be dependent on HER3 for proliferation, I was surprised to see that the nude EV20 antibody had no effect on cell number (Fig EV1D). They should show the HER3 and P-HER3 levels for all the control and the EV20 treated cell lines used in this experiment (Fig 2). The correlation between HER3 levels and response to EV20/MMAF in Fig 2E could also be done using P-HER3 levels. See also my general comment 2 above.

In Fig 2K there is still some growth of the 2 KD cell lines. It would be interesting to see what the levels of P-HER3 and P-AKT are in these cells. This might explain why they are not completely blocked.

Fig 3-

Since they are comparing EV/MMAF to trastuzumab, it would be interesting to see what happens to HER2 and HER3 amounts in trastuzumab treated parental and resistant cells (BTRH & BT474).

This might shed some insight into why the BTRH cells are resistant.

Fig 5-

The results showing that a single dose of 10mg/ml EV20/MMAF causes tumor regression and that even after 1 year there was no recurrence are very impressive. The x-axis of panel A should read days after treatment, or days of the experiment, since there was only 1 treatment dose.

They monitor EV20/MMAF in various organs of the mouse and in the tumor (Fig 5A and EV3A). They also monitor weight as a sign of general health (EV3C). This raises the question as to whether the EV20 antibody recognizes murine HER3. I know that it was raised against human HER3 ECD protein, but I could not find any information on whether it also recognizes the murine protein. Please clarify this since of course if it cannot bind the receptor on mouse organs, only on the human tumors, this would mean that it would be unlikely to have any side effects that might be relevant for treatment of humans.

In this section they also discuss an experiment with multiple dosing of 3.3mg/kg EV20/MMAF that results in full regression (Supp Fig 2B). I cannot find these data in the paper- please correct.

Fig 6-

In the final experiment they look at other resistant models - neratinib, lapatinib and T-DM1. These agents are also important to work with, but I had the impression that this section was hastily put together. At the minimum, I think it would be important to include the P-HER2, P-HER3 and P-AKT status in these models.

Discussion

In general, this section is fine, but some of the comments on HER3 levels and activity of EV20/MMAF should be rewritten in light of my general comment 2 above.

Minor comments:

- Spelling error on pg 10- tisular should read tissue.
- Supp fig 1D on pg 7 should be renamed EV 1D.

Referee #3 (Remarks for Author):

This manuscript describes efforts to apply a HER3 monoclonal conjugate that these authors have characterized and published in at least 3 previous studies as referenced by the authors to target HER2 tumor cells made resistant to different HER2 therapies. The initial characterization of the mouse erbB3/HER3 monoclonal (G. Sala et al, *Oncogene*, 2012) indicated that on its own, it inhibited signaling and caused degradation of erbB3 (HER3) in a breast tumor line, and inhibited growth of tumors. This humanized HER3 monoclonal as a drug conjugate was subsequently reported to abrogate receptor signaling and induce receptor downregulation and be rapidly internalized by tumor cells as well as inhibit growth of a variety of tumor types including prostate, ovarian and pancreatic cancers (G. Sala et al, *Translational Oncology*, 2013). The same humanized drug conjugated monoclonal was then characterized for effects including uptake and cell killing in melanoma (E. Capone et al, *J. of Controlled Release*, 2018). Unfortunately, there is little if any new information in the present study, some results contradict their own previous studies, and the resistance models are not sufficiently well developed to allow meaningful conclusions as indicated below:

1. The resistant clones or mass cultures are selected in tissue culture, but data concerning how resistant the "resistant" cells are is shown in only one case (Fig.1B). Moreover, the effects are not striking in that the authors observe only a 50% reduction in growth of the parental BT474 cells. Other models for resistance selection are not sufficiently well described to evaluate. Thus, the degree to which their cell systems accurately model in vivo resistance is a major concern. Clearly, if the authors' performed experiments with the toxin conjugated HER2 monoclonal as they indicate, they should be able to demonstrate a presumably high degree of resistance of the selected cells to cell killing by the same anti HER2 drug conjugate compared to the relative sensitivity of the parental tumor cells. Such experiments would be needed to demonstrate true drug resistance and

thus to determine whether such resistance provides a reasonable model for in vivo anti HER2 resistance mechanisms.

2. An alternative approach would be to use a model system in which resistance is selected in vivo. By this approach, the authors could establish the extent to which the effects of HER2 and HER3 monoclonals vs drug conjugated HER3 monoclonal impact in vivo growth of an in vivo tumor to Herceptin, lapatinib, or drug conjugated HER2 monoclonal. Such experiments would be critical to making meaningful conclusions concerning the ability of the HER3 drug conjugated monoclonal to overcome HER2 targeted tumor resistance.

3. This concern is reinforced by the lack of any meaningful differences in the responses of parental vs the authors' selected "resistant" tumor cell populations to the drug conjugated HER3 monoclonal. The authors detect no differences in HER2 or HER3 levels or phosphorylation in any of their attempts to generate HER2 therapy resistant breast tumor cells. This is particularly disturbing in view of their published studies that report that this is the case.

4. What is known about resistant mechanisms would be helpful as well as this would allow the authors to test whether any of their resistant lines exhibit such properties.

5. Since the growth stimulatory effects of HER3 for HER2 are well established, the lack of tumor growth inhibition in culture or in vivo by the HER3 monoclonal alone in the experiments presented would seem to argue that it has little or no biological effects on signaling by the heterodimer. Why this is the case is not clear as they have reported that the same monoclonal is growth inhibitory on its own against HER3 expressing tumors.

6. Finally, there is substantial evidence that some of the HER2 monoclonal's activity in vivo is mediated by immune mechanisms that can't be modeled in an immunocompromised mouse model used by the authors. Thus, a case could be made for testing the HER3 monoclonal and the drug conjugate in a mouse model of HER2/HER3 overexpression and drug resistance as this would allow them to explore this mechanism as well.

1st Revision - authors' response

4th Mar 2020

Referee #1 (Remarks for Author):

REFEREE'S COMMENTS:

The manuscript by Gandullo-Sanchez et al entitled "HER3 targeting with an antibody drug conjugate bypasses resistance to anti-HER2 therapies" describes a new study using EV20/MMAF, an anti-HER3 drug conjugate that displayed, in preclinical setting, a potent antitumor activity in cutaneous melanoma, to demonstrate its potent anti-tumor properties in the context of HER2+ breast cancers resistant to HER2 targeting therapies such as trastuzumab, lapatinib, neratinib. The authors explored the action of EV20/MMAF on in vitro primary and secondary (acquired) resistance models and reported a long-lasting tumor growth inhibition induced by EV20/MMAF on one mice model of trastuzumab-mediated resistance. From this point of view, the topic of the paper is clearly of interest as this study strengthens the arsenal of strategies targeting HER3 with very promising results in the field of breast cancers, and more generally validates and enlarges the anti-tumor potential of EV20/MMAF.

AUTHOR'S RESPONSE: We welcome the positive comments of the Reviewer about the paper, recognizing the importance of our study about the efficacy of EV20/MMAF in breast cancer.

REFEREE'S COMMENTS:

However, an important part of the manuscript is devoted to re-confirming mechanisms or properties related to either the antibody EV20 or the drug conjugate such as the HER3 dependent cell killing activity, antibody-mediated internalization or drug-mediated cell cycle termination, that have been described in previous publications. Even if these points are supported here by new data/techniques, they only confirm data.

AUTHOR'S RESPONSE: In other cellular models some data commented by the Reviewer have already been published. However, such data were not available in the case of our cellular models (both parental and resistant) and therefore these experiments had to be performed and presented.

REFeree'S COMMENTS:

It would have been more interesting to extend in vivo the in vitro studies on the efficacy of this ADC against different resistance models (primary resistance, drug and/or trastuzumab ADC). This would have brought even more validity and facilitate clinical translation.

AUTHOR'S RESPONSE: Following the indications of this Reviewer, as well as some comments of Reviewer 3, we have extended the in vivo and in vitro studies. We analyzed the action of EV20/MMAF in three additional models: (i) a second model of trastuzumab resistance (data now provided in Fig 5B); (ii) a model of in vivo resistance to the trastuzumab ADC T-DM1 (data now provided in Fig EV5D); and (iii) a model of primary resistance to trastuzumab (data now provided in Fig 5C). The first additional model is based on a clone (#10) of BT474 cells selected for its in vitro resistance to trastuzumab (see figure 1C). Tumors created in mice using that model retain resistance to trastuzumab in vivo (data provided in Figure 5B). The T-DM1 resistant model is also derived from BT474 cells and also retains in vivo resistance (see data now provided as Fig EV5D and our previously published report Rios-Luci et al Cancer Res. 2017 Sep 1;77(17):4639-4651). We also prepared a third in vivo model using HCC1954 cells which resulted primary insensitive to trastuzumab in vitro (see Fig 2D) and retained resistance to trastuzumab in vivo (data now provided in Fig 5C).

We observed strong antitumoral action of EV20/MMAF in all three new models.

We also provide additional data on the models of resistance used, including a more detailed description of their generation (in the materials and methods section), dose-response curves of their sensitivity to the treatments against which they were made resistant (Fig EV1A, Fig EV5A, Fig EV5B, Fig EV5C), as well as levels of HER2, HER3, pHER2, pAKT and pHER3 (all in Fig 6 westerns).

REFeree'S COMMENTS:

The experiments are reasonably well designed and most results are sound.

However, the manuscript needs to be improved by the following recommendations:

Specific comments

1- In the introduction, the authors give only very general information about HER2-positive breast cancers and HER3. The role of HER3 in treatment failure in cancer therapy is well known and different approaches targeting HER3 are currently ongoing (pre-clinic and clinic) in different cancers. A more precise contextualization would have been a plus, particularly with a brief overview of the current researches in the field. As well, a brief reminder of the properties of EV20, notably its capacity to block ligand dependent and independent activation of HER3 would be of help.

AUTHOR'S RESPONSE: Following the comments of the Reviewer, we have now added several paragraphs (in the introduction as well as in the discussion sections) about the role of HER3 in treatment failure, strategies to target it and clinical results. We have also included comments on the properties of EV20, notably its capacity to block ligand dependent and independent activation of HER3.

REFeree'S COMMENTS:

2- Concerning the model of secondary resistance, very little technical details are given, neither in the text nor in Fig1A that does not provide much and thus can be removed.

- What low density of cells really means (Fig.1A legend)?

- Frequency of trastuzumab replenishment during the 1-year treatment?

- Is resistance assessed only by viability assay? No phenotype investigation?

- How are the cells maintained once made resistant, just in medium without low dose of trastuzumab? Is the resistant phenotype stable? A reference or more technical details should be given.

AUTHOR'S RESPONSE: Figure 1A was prepared to schematize the generation of trastuzumab-resistant cells and also to illustrate how we established the criteria for primary resistance based on the data generated using the secondary resistant cells. We acknowledge that this is a mere scheme, and if the Reviewer insists, we can delete it or move it to the extended view section.

Most of the characteristics of the secondary trastuzumab-resistance model are detailed in a recent paper published by our laboratory (Cancer Lett. 2020 Feb 1;470:161-169), not available for citation when we first submitted the actual paper, but that is now cited. In addition, we now provide more details about the different resistant models. First, dose-response curves of resistance to the drugs against which the different resistant models were created are now provided in Fig EV1A (for

trastuzumab) and Fig EV5A-C (for T-DM1, neratinib and lapatinib). Second, we expanded the materials and methods section to include additional details about how we generated the resistant cells and have also included some references. In the specific case of the generation of the trastuzumab-resistant BT474 cells we describe that parental cells were plated at 5,000 cells/150 mm dish and media containing trastuzumab (50 nM) was changed once weekly. This information is now also provided in the figure legend.

Resistance in all models was assessed by cell counting or MTT assays. Phenotypically, the resistant cells very much look like the parental BT474 cells. Moreover, most of the biochemical studies we performed show little differences among resistant and sensitive cells. With respect to how are cells maintained once made resistant after one year of treatment, we have observed that resistance to trastuzumab is preserved even in the absence of continuous pressure by the drug. The phenotype is therefore stable (please consult *Cancer Lett.* 2020 Feb 1;470:161-169).

REFEREE'S COMMENTS:

3- The authors stated that resistance to the action of trastuzumab was not caused by a loss of phosphorylated HER2, but it is not so obvious from the western blot (Fig 1C). Addition of intensity values obtained by densitometry could be of interest. As well in the same blot, an increase of HER3 is observable but is not commented.

AUTHOR'S RESPONSE: Following the Reviewer's indications, we have incorporated intensity values to the Westerns shown in Figure 1C and also to other Westerns included in the paper (those whose quantitation was not originally included).

Regarding whether to comment on the increase in HER3 seen in the figure, we don't want to make a strong point about that, since we feel that such differences are subtle. In fact, in figure 2E, the HER3 levels in BTRH are very similar to those of BT474 cells.

REFEREE'S COMMENTS:

4- In the legend of Fig1, does the sentence "Data are represented as mean + SD of triplicate..." apply to all cell counting graphs? If not, the number of performed experiments should be added. The error bars are missing or not visible for many points. These remarks are valid for most figures.

AUTHOR'S RESPONSE: Cell counting experiments have been performed in triplicate and at least twice. We have now made this clear in the figures. The reason by which error bars are not seen in some points is because they are very small and are therefore covered by the symbols used in the graphics. That now appears explained in the figure legends.

REFEREE'S COMMENTS:

5- In Fig2B the HER2 level is lower than that of HER3 in BT474 and BTRH while in Fig.1C it is the opposite. Is there any explanation?

AUTHOR'S RESPONSE: Please note that the HER2 level in Fig. 2B is higher than that of HER3. In Fig. 1C it is true that the image showing the signal of HER3 is stronger than that of the one showing the HER2 signal. That is a gel exposure issue.

REFEREE'S COMMENTS:

6- In Fig2F, is there any explanation for the lower cytotoxic activity of EV20/MMAF on HCC1569 despite a similar level of HER3, and compared to its activity on HCC1954?

AUTHOR'S RESPONSE: This is a very good observation on which we have also reflected. In fact, HER3 levels of HCC1954 are lower than those of HCC1569 and yet the former are more sensitive to the action of EV20/MMAF. As we discuss in the paper, HER3 expression levels is a factor that defines sensitivity to EV20/MMAF but since there is no a strict correlation between HER3 levels and IC₅₀ values, other factors must contribute to the sensitivity to the ADC. It would not be surprising that internalization rate of HER3 upon binding to EV20/MMAF or lysosomal proteolytic activity could also represent important rate-limiting steps in the action of the ADC.

REFEREE'S COMMENTS:

7- The paragraph (and the number of panels Fig.2G-K) on the HER3 dependent cell killing activity of EV20/MMA (p7) should be condensed to the data considered by the authors as the most relevant and new in relation to the data already published on this subject. The other data can be moved to supplementary data.

The linear regression analysis is very briefly described, so a more descriptive figure showing a plot with all data and the linear regression analysis should be given instead of Fig.2C.

AUTHOR'S RESPONSE: Following the indication of the Reviewer, we moved former figures 2G, 2H, 2I, and 5B to expanded view panels. The linear regression analyses are now more detailed, and we have included the blots from which they were created, together with the linear regression data, to a full expanded view figure (EV2). Also, to avoid condensation of the main figures, most of the novel data have been incorporated as EV panels.

REFEREE'S COMMENTS:

8- The sentence "For these experiment, cells were incubated...." should be removed (experimental details given in Mat & Met).

The name of the cells used in the internalization of the biotinylated ADC is not mentioned in the text, only in the figure.

AUTHOR'S RESPONSE: The sentence to which the Reviewer refers to has been removed. The cell lines in which the internalization experiments using biotinylated ADC were performed are now also described in the text.

REFEREE'S COMMENTS:

9- What about the impact of HER3 ligand on the cytotoxic activity of EV20/MMAF? it would have been of interest to verify (as they did for the internalization, cell-cycle..), that the presence of neuregulin does not affect the cytotoxicity of EV20/MMAF.

AUTHOR'S RESPONSE: We have performed experiments to explore whether the HER3 ligand neuregulin affected the cytotoxicity of EV20/MMAF. The results indicate that the presence of the ligand does not affect the activity of EV20/MMAF. These results are now described in the text, and shown in Fig EV1H.

REFEREE'S COMMENTS:

10- It is not clear whether the mice to be euthanized were picked in the same group or among the 3 treated groups (p10)?

Fig.5B is of poor quality: the samples are barely visible due to truncation of the gel and the loading control varies a lot (both in amount and number of band). How was made the quantification based on this? As well, in Fig EV3A, variability of the loading control: in this respect, one could assume that the level was also high in kidney and intestine, no?

AUTHOR'S RESPONSE: The experiment to assess the tissue distribution of EV20/MMAF was performed on one group of untreated mice bearing BTRH tumors. They were treated with EV20/MMAF and then euthanized at 24 hours or after two weeks. This is now detailed in the text. We used the same amount of protein (500 micrograms, now detailed in the materials and methods section as well as in the figure legend) of each tissue or tumor to perform immunoprecipitation of EV20/MMAF in the samples shown in figure 5B and former Fig EV3A (now Figs EV4A and B). Also, equal amounts of protein (50 μ g) from the different tissues were run in parallel gels that were analyzed for calnexin. As the Reviewer noticed, the amount/bands of calnexin varied among tissues, and we acknowledge that this may create confusion to the reader. For that reason, we prefer to delete the calnexin Westerns.

With respect to the quantitation question: the signal in each tissue of the untreated animals was subtracted from the signal obtained in the treated animals. This is now detailed in the figure legend.

REFEREE'S COMMENTS:

11- During the in vivo studies, do the BTRH cells remain resistant to trastuzumab? A control with trastuzumab treatment is missing and should be added to ensure the stability of the phenotype. Or at least it should be discussed.

As well, does an ADC-control be of interest?

AUTHOR'S RESPONSE: It is necessary to explain that we observe two types of in vivo behaviors in BT474 cells made resistant to trastuzumab in vitro.

One of them is exemplified by BTRH#10 cells, resistant to the action of trastuzumab both in vitro (Fig 1C) and in vivo (data now provided as Fig 5B). As can be seen in that figure, tumors created in mice injected with these cells are sensitive to the antitumoral action of EV20/MMAF.

The second type of behavior is represented by BTRH cells, also resistant to trastuzumab in vitro. However, tumors created in mice injected with these cells initially respond to trastuzumab in vivo and then relapse, becoming resistant in vivo (now commented in the results section of the paper). We tested the action of EV20/MMAF on three such BTRH-derived tumors that became resistant in vivo to weekly trastuzumab. These tumors made resistant in vivo to the action of trastuzumab

responded to EV20/MMAF, as shown in the figure below, which is provided for Reviewer's inspection and not for publication. If the Reviewers consider that the data is important and should be part of the manuscript, we will incorporate them.

The figure illustrates the action of trastuzumab on three tumors of different sizes, created by injecting BTRH cells in the mammary fat pads of female nude mice. Trastuzumab initially provoked tumor regression, but tumors relapsed over time, even though trastuzumab was administered weekly. Mice with relapsed tumors were then treated with EV20/MMAF at 3.3 mg/kg, where indicated. The two different *in vivo* behaviors of tumors from BTRH and BT474#10 cells with respect to their sensitivity to trastuzumab very much suggests that they must use different resistance mechanisms to become refractory to trastuzumab. Furthermore, the fact that EV20/MMAF caused tumor regression in both models (actually, in all the models tested) adds value to our findings reinforcing the hypothesis that such drug may be useful in the fighting of HER2+ tumors, which become refractory to trastuzumab.

REFEREE'S COMMENTS:

12- The discussion is very focused on the results without much openness to the literature. It should be more detailed to highlight the advantage or/and weakness of their molecule compared to the various molecules in pre-clinical and clinical trials in progress (for example anti-HER3 antibody drug conjugate U3-1402?)

AUTHOR'S RESPONSE: Several paragraphs addressing the advantages of EV20/MMAF and other anti-HER3 ADCs with respect to nude antibodies, the potential clinical applications of HER3 ADCs, and the clinical opportunities for EV20/MMAF have been included. We wish to point out that comparison of EV20/MMAF with available anti-HER3 ADCs has not been published. Therefore, it is difficult to predict which ADC could be better from the clinical point of view. Yet, it is worth mentioning that second generation ADCs targeting HER2 (e.g. trastuzumab-deruxtecan) proved clinically efficacious in patients which developed resistance to several previous anti-HER2 therapies, including T-DM1. This relevant clinical observation (now commented and referenced in the text) exemplifies the value of targeting the same cell surface molecule with ADCs bearing different payloads, as could be the case of EV20/MMAF and U3-1402. That identifies a clear potential benefit of having EV20/MMAF added to the repertoire of anti-HER3 drugs.

REFEREE'S COMMENTS:

13- Mat et Met section should be improved so that the experiments can be reproduced because several methodological details are missing (number of cells, secondary antibodies for detection of tubulin and LAMP1 in indirect IF, number of replicate, p-Tyr antibodies which ones for HER2 and HER3? ..)

AUTHOR'S RESPONSE: We have increased the information on the experimental procedures in the materials and methods section, and carefully inserted references detailing some experimental

procedures. Dilutions of antibodies are now included as well as their catalogue number, when commercial, in an Appendix Table.

For immunofluorescence, the number of BT474 or BTRH cells plated was 100,000-150,000 cells in each well of a 35 mm dish which containing a square coverslip. The secondary antibodies used for detection of tubulin (Cy2-labeled goat anti-mouse IgG from GE Healthcare Life Sciences) or LAMP1 (Cy2-conjugated anti-rabbit from Jackson ImmunoResearch) are now detailed. In general, the in vitro experiments were repeated at least two times. We have attempted to specify that in the figure legends. The p-Tyr, pHER2 and pHER3 are described in the reagents section of the materials and methods.

REFEREE'S COMMENTS:

When do they use anti-EGFR antibody for immunoprecipitation?

Anti-HER3 has been described: do the authors mean the EV20?

Can the authors explain the choice not to give statistical analyses of their data?

AUTHOR'S RESPONSE: In fact, we do not present data about EGFR. Accordingly, we have deleted the sentence about this antibody in the materials and methods section.

Details on the HER3 antibody used for Western (it is not the EV20) are now given.

Initially, we did not feel necessary to include statistical analyses in graphics in which the effect (or lack of it) was clear. We have now included statistical analyses for graphs in which we felt those analyses could help. The exact *P* values have been incorporated as Appendix tables.

REFEREE'S COMMENTS:

14- The figures are very dense and could be reduced by selecting the most relevant data most of the time.

Addition of intensity values obtained by densitometry should be added to all western blot images

The error bars are often not visible in the proliferation curves

The color legend of the immunofluorescence images in Fig. 3A and D are missing in the legend.

AUTHOR'S RESPONSE: Some panels of main figures have been moved to the EV data. We have taken care also not to increase figure density due to the incorporation of novel data, which has been mostly placed as EV panels.

Blots not formerly quantitated, have now been quantitated and the intensity values added.

As mentioned elsewhere in this letter, error bars are sometimes so small that cannot be seen as they are covered by the symbols of the graphics. That is explained now in the figures.

What each color corresponds to is shown now in the legend of Figure 3.

REFEREE'S COMMENTS:

Minor modifications:

p7 supplementary Fig1D should be replaced by Fig. EV1D

p10, Twenty instead of 24, no number at the beginning of a sentence

p11, supplementary Fig 2B should be replaced by Fig. EV3B

AUTHOR'S RESPONSE: These three corrections have been made.

Referee #2 (Comments on Novelty/Model System for Author):

REFEREE'S COMMENTS:

I rated the novelty medium since one of the results on the MOA of the ADC has already been shown for the humanized antibody. I mentioned this in the review. Otherwise the rest is novel. So it's only a small effect on novelty.

AUTHOR'S RESPONSE: The Reviewer recognizes the novelty of our studies. Probably, the most impacting aspect of the present work is to have found a clinical application for the HER3 ADC for tumors that become resistant to conventional therapies that are used in patients with HER2 positive tumors.

Referee #2 (Remarks for Author):

REFEREE'S COMMENTS:

The manuscript by Gandullo-Sanchez and collaborators presents cellular and in vivo work on a novel HER3 targeting antibody drug conjugate (ADC), EV20/MMAF. The ADC is composed of a humanized antibody targeting HER3, EV20, conjugated to auristatin F, a tubulin polymerization-

blocking compound. There are two important reasons to generate and study therapeutics targeting HER3. First, since the approval of trastuzumab for treatment of HER2 positive breast cancer, many, but not all patients with high HER2 levels have benefitted from treatment. It is well known that at least some of the trastuzumab resistance is due to activation of HER3 the main regulator of the PI3K/AKT pathway. Second, oncogenic mutations in HER3 have been found in many cancer types, most prevalently in gastric and colon cancers. The work presented by Gandullo-Sanchez et al concentrates on trastuzumab resistant models. The paper is clearly written and the presented data are interesting and well discussed. I do have a few suggestions and criticisms, which if addressed properly could make the paper even better.

AUTHOR'S RESPONSE: We appreciate these positive comments from this Reviewer about our work. We have thought about the possibility of including a small paragraph in the discussion section about HER3 mutations in cancer and the possibility of targeting oncogenically mutated HER3 with ADCs as EV20/MMAF. We have done that and included a recent reference on HER3 mutations in cancer.

REFEREE'S COMMENTS:

General comments:

1. There are now a number of HER3 targeting therapeutics that are in clinical development. The clinicaltrials.gov site lists at least 25 trials with antibodies targeting HER3 in various types of human cancer. There are even publications describing clinical trial results for some of them, e.g., lumretuzumab. I was surprised that there was no discussion of at least some of these in the manuscript. It would be important to discuss first, why the new EX20/MMAF ADC might be better than what's already available and second, whether its MOA differs and might have some advantages compared to others.

AUTHOR'S RESPONSE: As mentioned above (see response to Reviewer 1, point 12) several paragraphs addressing the advantages of EV20/MMAF and other anti-HER3 ADCs with respect to nude antibodies, the potential clinical applications of HER3 ADCs, the advantage of having different payloads in ADCs targeting HER3 and the clinical opportunities for EV20/MMAF have been included. References have also been included.

REFEREE'S COMMENTS:

2. Throughout the paper, the authors seem to equate HER3 levels with oncogenic activity, arguing, e.g., that the human stromal cell line HS5 has low HER3, which could explain why it is not sensitive to the antibody (Fig 2). In my opinion, they should be looking at P-HER3 levels, since this is a better sign of activity and dependency. There is one P-HER3 blot shown, but I would like to see more, particularly in the *in vivo* work in Fig 5.

AUTHOR'S RESPONSE: Actually, in our work we have not placed special emphasis on the oncogenic effect of HER3. Rather, we take advantage of the high expression of HER3 in resistant tumors to use that membrane protein as an EV20/MMAF anchor site. That is why we have not insisted on defining pHER3 levels throughout this work. However, we understand that for cells in which HER3 may exert a prooncogenic action, its targeting with EV20/MMAF may sum to the action of the ADC as a transport vehicle to deliver the cytotoxic payload. We now comment that in the discussion section of the paper. Moreover, we have now included more Western blot studies of pHER3 levels throughout the paper, including those requested in the *in vivo* work of Fig 5.

REFEREE'S COMMENTS:

3. The original murine mAb MP-RM-1 was described and studied in Sala et al 2012 *Oncogene*, and the humanized version EV20 was characterized in Sala et al 2013 *Translational Oncol*. In both these papers it was shown that the antibodies downregulate HER3 levels and AKT activity (p-AKT). Using microscopy, it was shown in Sala et al 2013 that there was a time-dependent increase in co-localization of EV20/LysoTracker and that the antibody was still bound to HER3. This work impacts somewhat on the originality of the current paper, where they also nicely show with stills and with videos that EV20/MMAF tracks to the lysosomes.

AUTHOR'S RESPONSE: As the Reviewer well mentions, there are data about the vehiculization of EV20/MMAF towards the lysosomes in other cellular models. However, such data were not available in the case of our cellular models (both parental and resistant) and therefore these experiments had to be performed, and that is the reason why they are shown.

REFeree'S COMMENTS:

4. The final general comment/criticism that I have relates to the *in vivo* activity of EV20/MMAF. Both previous papers show that the MP-RM-1 and the EV20 antibodies block *in vivo* outgrowth of different human xenograft models. In the current manuscript, they show that "the nude antibody EV20 was unable to prevent tumor growth" (Fig 5A, pg 10). I was surprised to find that there was no discussion of the difference between their results and what has been published.

AUTHOR'S RESPONSE: The Reviewer is right. Nude EV20 has been reported to have an effect on tumor growth in some *in vivo* models. Moreover, in one of the models we now include (the HCC1954 model), EV20 had an effect, stopping tumor growth. However, in the case of the BT474-derived models, we were unable to see an appreciable effect of single-dose EV20. It is possible that such dosing schedule is responsible for the lack of effect of EV20 in BT474-derived models, since former studies performed in other models were done using EV20 twice per week, for periods of 4-5 weeks. Moreover, previously published data reported the effect of EV20 administered to animals with tumors substantially smaller than those in the present study (100 mm³ vs 500 mm³). These circumstances are now commented in the discussion section of the paper. The reason for doing single-dose EV20 was to compare its effectiveness with single-dose EV20/MMAF.

REFeree'S COMMENTS:

Specific comments to the figures:

Fig 2 -

Since SKBR3 and BT474 are known to be dependent on HER3 for proliferation, I was surprised to see that the nude EV20 antibody had no effect on cell number (Fig EV1D). They should show the HER3 and P-HER3 levels for all the control and the EV20 treated cell lines used in this experiment (Fig 2). The correlation between HER3 levels and response to EV20/MMAF in Fig 2E could also be done using P-HER3 levels. See also my general comment 2 above.

Levels of HER3 and pHER3 of all the cell lines used in figure 2 have now been included. Data on correlation studies between pHER3 and EV20/MMAF is also now provided in Fig EV2E and Fig EV2F.

REFeree'S COMMENTS:

In Fig 2K there is still some growth of the 2 KD cell lines. It would be interesting to see what the levels of P-HER3 and P-AKT are in these cells. This might explain why they are not completely blocked.

AUTHOR'S RESPONSE: Fortunately, we still had frozen extracts from that experiment, and could therefore perform the pHER3 and pAKT Westerns, which are now provided (now is Fig 2G). Levels of pHER3 and pAKT are decreased by the KD, but not absent and this may explain the partial inhibitory effect of the KD, as the Reviewer indicates.

REFeree'S COMMENTS:

Fig 3-

Since they are comparing EV/MMAF to trastuzumab, it would be interesting to see what happens to HER2 and HER3 amounts in trastuzumab treated parental and resistant cells (BTRH & BT474).

This might shed some insight into why the BTRH cells are resistant.

AUTHOR'S RESPONSE: The effect of trastuzumab on HER2 and HER3 amounts in BT474 and BTRH cells is now included (Fig EV3C and Fig EV3D).

REFeree'S COMMENTS:

Fig 5-

The results showing that a single dose of 10mg/ml EV20/MMAF causes tumor regression and that even after 1 year there was no recurrence are very impressive. The x-axis of panel A should read days after treatment, or days of the experiment, since there was only 1 treatment dose.

AUTHOR'S RESPONSE: We have now replaced "Days of treatment" for "Days after treatment".

REFeree'S COMMENTS:

They monitor EV20/MMAF in various organs of the mouse and in the tumor (Fig 5A and EV3A). They also monitor weight as a sign of general health (EV3C). This raises the question as to whether the EV20 antibody recognizes murine HER3. I know that it was raised against human HER3 ECD protein, but I could not find any information on whether it also recognizes the murine protein. Please clarify this since of course if it cannot bind the receptor on mouse organs, only on the human

tumors, this would mean that it would be unlikely to have any side effects that might be relevant for treatment of humans.

AUTHOR'S RESPONSE: EV20 does not recognize rodent HER3 and this was formerly been published (Capone, E., et al., EV20-Sap, a novel anti-HER-3 antibody-drug conjugate, displays promising antitumor activity in melanoma. *Oncotarget*, 2017. 8(56): p. 95412-95424). That fact, together with its potential impact in the evaluation of the toxicity of EV20/MMAF, are now commented and cited in the results section of the paper.

REFEREE'S COMMENTS:

In this section they also discuss an experiment with multiple dosing of 3.3mg/kg EV20/MMAF that results in full regression (Supp Fig 2B). I cannot find these data in the paper- please correct.

AUTHOR'S RESPONSE: The Reviewer is right. The illustration representing the data with multiple dosing of 3.3 mg/kg was erroneously indicated as Supplementary figure 2B (now Fig EV4C). We have corrected that in the main text.

REFEREE'S COMMENTS:

Fig 6-

In the final experiment they look at other resistant models - neratinib, lapatinib and T-DM1. These agents are also important to work with, but I had the impression that this section was hastily put together. At the minimum, I think it would be important to include the P-HER2, P-HER3 and P-AKT status in these models.

AUTHOR'S RESPONSE: We believe that the inclusion of these resistance models increases the value of our study, as it extends the applicability of EV20 / MMAF to combat resistance to other approved anti-HER2 therapies, which are used even after resistance to trastuzumab. Levels of pHER2, pHER3 and pAKT are now included in the figure, as requested, and mentioned in the text.

REFEREE'S COMMENTS:

Discussion

In general, this section is fine, but some of the comments on HER3 levels and activity of EV20/MMAF should be rewritten in light of my general comment 2 above.

AUTHOR'S RESPONSE: Some comments on this have been added. We have also commented that for cells in which HER3 function is important for growth, its targeting and down regulation of pHER3 may add to the action of the ADC.

REFEREE'S COMMENTS:

Minor comments:

-Spelling error on pg 10- tisular should read tissue.

-Supp fig 1D on pg 7 should be renamed EV 1D.

AUTHOR'S RESPONSE: We have modified the text accordingly.

Referee #3 (Remarks for Author):

REFEREE'S COMMENTS:

This manuscript describes efforts to apply a HER3 monoclonal conjugate that these authors have characterized and published in at least 3 previous studies as referenced by the authors to target HER2 tumor cells made resistant to different HER2 therapies. The initial characterization of the mouse erbB3/HER3 monoclonal (G. Sala et al, *Oncogene*, 2012) indicated that on its own, it inhibited signaling and caused degradation of erbB3 (HER3) in a breast tumor line, and inhibited growth of tumors. This humanized HER3 monoclonal as a drug conjugate was subsequently reported to abrogate receptor signaling and induce receptor downregulation and be rapidly internalized by tumor cells as well as inhibit growth of a variety of tumor types including prostate, ovarian and pancreatic cancers (G. Sala et al, *Translational Oncology*, 2013). The same humanized drug conjugated monoclonal was then characterized for effects including uptake and cell killing in melanoma (E. Capone et al, *J. of Controlled Release*, 2018). Unfortunately, there is little if any new information in the present study, some results contradict their own previous studies, and the resistance models are not sufficiently well developed to allow meaningful conclusions as indicated below:

AUTHOR'S RESPONSE: This Reviewer knows our previous work on EV20/MMAF and the nude antibody. While some information about this antibody has been published, as this Reviewer and

Reviewer 2 mention, its potential therapeutic value in HER2 resistances has only been explored here. In that respect, the study is novel, as also indicated by the Reviewers. With respect to the comment that some of the present results contradict previous ones, we need more precise information about which are those ones. In case the Reviewer refers to the lack of action of the nude antibody in our models, please see our answer to that in other parts of this response letter. Finally, with respect to the resistance models, we have extended the information provided, as we have commented below and elsewhere (some answers to Reviewers 2 and 3).

REFEREE'S COMMENTS:

1. The resistant clones or mass cultures are selected in tissue culture, but data concerning how resistant the "resistant" cells are is shown in only one case (Fig.1B). Moreover, the effects are not striking in that the authors observe only a 50% reduction in growth of the parental BT474 cells. Other models for resistance selection are not sufficiently well described to evaluate. Thus, the degree to which their cell systems accurately model in vivo resistance is a major concern. Clearly, if the authors' performed experiments with the toxin conjugated HER2 monoclonal as they indicate, they should be able to demonstrate a presumably high degree of resistance of the selected cells to cell killing by the same anti HER2 drug conjugate compared to the relative sensitivity of the parental tumor cells. Such experiments would be needed to demonstrate true drug resistance and thus to determine whether such resistance provides a reasonable model for in vivo anti HER2 resistance mechanisms.

AUTHOR'S RESPONSE: We agree that the information about the different resistant models was very scarce in the former version of the paper. Because of space limitations, we were very concise in their description. Now we include more information, including dose-response studies to describe how resistant those models are. Papers detailing trastuzumab or T-DM1 resistant models are now referenced in the materials and methods section and more details are also provided in that section. A short paragraph describing the generation of BT474 cells resistant to lapatinib or neratinib is also included. In addition, dose-response analyses in which we show resistance of different cells to trastuzumab (Fig EV1A), neratinib (Fig EV5A), lapatinib (Fig EV5B) or T-DM1 (Fig EV5C), are now provided. With respect to the resistance model to T-DM1, which is specifically mentioned by the Reviewer, and in addition to the data provided now (Fig EV5C), please consult our 2017 report in which these cells were described (Rios-Luci et al, Cancer Res. 77:4639).

Regarding the effect of trastuzumab on parental BT474 cells, the % inhibition achieved by treating them with trastuzumab varies depending on the time of exposure to that drug. That is not surprising since the effect of trastuzumab is cytostatic rather than cytotoxic, and the longer the experiment the larger the % inhibition. In fact, in our hands, the % inhibition ranges from 50-75% in a 5-10 days window time. Even with ample incubation times, we never obtain 100% inhibition, and this fact indicates the presence of resistant cells as well, a circumstance that we and others harnessed to generate the trastuzumab-resistant cells.

REFEREE'S COMMENTS:

2. An alternative approach would be to use a model system in which resistance is selected in vivo. By this approach, the authors could establish the extent to which the effects of HER2 and HER3 monoclonals vs drug conjugated HER3 monoclonal impact in vivo growth of an in vivo tumor to Herceptin, lapatinib, or drug conjugated HER2 monoclonal. Such experiments would be critical to making meaningful conclusions concerning the ability of the HER3 drug conjugated monoclonal to overcome HER2 targeted tumor resistance.

AUTHOR'S RESPONSE: In this revised version, we provide three additional in vivo studies: (i) an in vivo study using BTRH#10 cells (a BT474 trastuzumab-resistant clone different from BTRH); (ii) an in vivo study using BT-TDM1R#6, which are resistant to the action of T-DM1; (iii) an in vivo model of primary resistance using HCC1954 cells. BTRH#10 and HCC1954 show both in vitro (Fig 1B, Fig EV1A and Fig 2D) and in vivo (Fig 5B and C) resistance to trastuzumab. BT-TDM1R#6 are resistant to T-DM1 (Rios-Luci et al, Cancer Res. 77:4639 (2017) and the data provided in Fig EV5C and Fig EV5D). Yet, as can be seen in Fig 5B, Fig 5C and Fig EV5D, in all these models EV20/MMAF caused tumor regression.

In our response to Reviewer 1, point 11, we describe how BTRH cells respond to trastuzumab in vivo.

REFEREE'S COMMENTS:

3. This concern is reinforced by the lack of any meaningful differences in the responses of parental vs the authors' selected "resistant" tumor cell populations to the drug conjugated HER3 monoclonal.

The authors detect no differences in HER2 or HER3 levels or phosphorylation in any of their attempts to generate HER2 therapy resistant breast tumor cells. This is particularly disturbing in view of their published studies that report that this is the case.

AUTHOR'S RESPONSE: We in fact found that EV20/MMAF exerted a profound antitumoral action in several models of resistance to drugs used in the therapy of HER2 tumors. That fact reinforces the hypothesis that EV20/MMAF may be useful in the fighting of HER2+ tumors that become refractory to those anti-HER2 therapies.

Regarding the effect of EV20/MMAF on the HER2 and HER3 levels and phosphorylation, please note that we provided evidence in Fig 3E and Fig 3F that EV20/MMAF decreased HER3 levels in a similar way in BT474 and BTRH cells.

With respect to levels or phosphorylation of HER2 and HER3, we now provide more data on the tyrosine phosphorylation these proteins in the different resistant models (Fig 1, Fig 2 and Fig 6). Some differences in HER2 phosphorylation are present.

Here, we consider worth sharing our experience during the generation of models resistant to anti-HER2 therapies used in the clinic. In all the models that we generated and analyzed, we failed to isolate pools/clones lacking HER2 and differences in total HER2 are subtle. Therefore, we could not focus on HER2 levels as responsible for the resistant phenotypes. With respect to the tyrosine phosphorylation levels, there are some differences, even though there is always resting tyrosine phosphorylation of HER2. That also makes us to be cautious with respect to the impact of those changes in the resistant phenotype.

REFEREE'S COMMENTS:

4. What is known about resistant mechanisms would be helpful as well as this would allow the authors to test whether any of their resistant lines exhibit such properties.

AUTHOR'S RESPONSE: This is an important observation. In fact, a substantial part of the work in our laboratory aims at deciphering the mechanisms of resistance to the clinical anti-HER2 therapeutic agents. We of course evaluated some established mechanisms, such as lack of HER2 or activation of PI3K, to cite some. As mentioned above, lack of HER2/pHER2 cannot explain the resistant phenotypes. Moreover, we never detected extreme changes in the PI3K/mTOR route in resistant cells as compared to the parental ones that could explain the resistant phenotype. It is worth mentioning that some papers investigating mechanisms of resistance to T-DM1 or trastuzumab have been published by our group. In a 2017 paper (Rios-Luci et al, Cancer Res. 77:4639), we showed that a decrease in lysosomal proteolytic activity could create resistance to the action of T-DM1. In two more recent reports (published in 2019 and 2020, referenced now in the paper) published in Cancer Letters we show that trastuzumab-resistant cells develop alterations in cell death mechanisms that make them resistant to the action of lapatinib or neratinib. In addition to these findings, these papers include biochemical and transcriptomic characterization of the resistant models generated. We still have work to do. For example, detailed characterization of the neratinib-resistant model is ongoing and we expect to publish our data on that in the near future.

REFEREE'S COMMENTS:

5. Since the growth stimulatory effects of HER3 for HER2 are well established, the lack of tumor growth inhibition in culture or in vivo by the HER3 monoclonal alone in the experiments presented would seem to argue that it has little or no biological effects on signaling by the heterodimer. Why this is the case is not clear as they have reported that the same monoclonal is growth inhibitory on its own against HER3 expressing tumors.

AUTHOR'S RESPONSE: In the HER2+ models used in present study, EV20 did not affect their in vitro proliferation, even though it was able to downregulate HER3 and pHER3 in some cell lines (data now provided as Fig EV1F and Fig EV1G). Stable dimeric interactions between HER2 and HER3 are difficult to detect in HER2 overexpressing cell lines such as BT474 or SKBR3 cells unless the HER3 ligand neuregulin is present (please see Sanchez-Martin Int J Cancer, 131:244 (2012). None of the cell lines studied here express neuregulin (our unpublished data, available upon request).

With respect to the in vivo growth inhibitory properties of the monoclonal, nude EV20 has been reported to have an effect on tumor growth in some in vivo models. In fact, in one of the models we now include (the HCC1954 model), it had an effect, stopping tumor growth. That effect contrasts with the lack of in vitro effect of EV20 on HCC1954. The reason for such discrepant behavior may lie in the physiological versus in vitro models and this is now discussed in the paper. In BT474-derived models no in vivo effect of single-dose EV20 was seen. It is possible that the single-dosing schedule is responsible for the lack of effect of EV20 in the BT474-derived models, since former in

vivo studies performed in other models were done using EV20 twice per week, for 4-5 weeks. The paper includes discussion on that.

REFEREE'S COMMENTS:

6. Finally, there is substantial evidence that some of the HER2 monoclonal's activity in vivo is mediated by immune mechanisms that can't be modeled in an immunocompromised mouse model used by the authors. Thus, a case could be made for testing the HER3 monoclonal and the drug conjugate in a mouse model of HER2/HER3 overexpression and drug resistance as this would allow them to explore this mechanism as well.

AUTHOR'S RESPONSE: We understand the relevance of antibody-dependent cellular cytotoxicity (ADCC) in the action of antibodies such as trastuzumab. However, as the Reviewer mentions, this would require the use of additional non-immunocompromised animal models. The fact that even in the immunocompromised mice EV20/MMAF exerted a strong antitumoral effect adds value to our hypothesis and findings supporting the exploration of that antibody to fight resistance to anti-HER2 therapies.

2nd Editorial Decision

20th Mar 2020

Thank you for the submission of your revised manuscript to EMBO Molecular Medicine. We have now received the enclosed report from the referee who reviewed the new version of your manuscript. As you will see, this referee is now supportive of publication, and I am thus pleased to inform you that we will be able to accept your manuscript pending the following final minor revisions and editorial amendments.

***** Reviewer's comments *****

Referee #2 (Remarks for Author):

The manuscript HER3 targeting with an Antibody Drug Conjugate Bypasses Resistance to anti-HER2 therapies by Gandullo-Sanchez et al has been greatly improved by the revision. Many of the reviewers' requests were followed and quite a bit of new data was added to the paper. I have only a few minor suggestions/questions.

Since there are many data entries in the paper and in the supplemental figures, it would be good for all the important information to be concisely presented in each figure legend. As an example, for the Fig 6 legend please write the name of the cell line in the legend and not just in the text.

For fig 5D legend, which mouse model is shown. From the text it is written BTRH, please add to the legend. Same is true for Fig 5E legend.

Please check all the other legends for clarity.

In the letter they added a figure to answer question 11 of reviewer 1 regarding trastuzumab resistance. I think this could be added to the paper as an additional EV figure. It shows interesting and important results.

2nd Revision - authors' response

24th Mar 2020

The authors performed the requested editorial changes.

Corresponding Author Name: Atanasio Pandiella

Manuscript Number: EMM-2019-11498-V2